# Differential processing of decision information in subregions of rodent medial prefrontal cortex

Geoffrey W Diehl, A David Redish*

Department of Neuroscience, University of Minnesota, Minneapolis, United States

**Abstract** Decision-making involves multiple cognitive processes requiring different aspects of information about the situation at hand. The rodent medial prefrontal cortex (mPFC) has been hypothesized to be central to these abilities. Functional studies have sought to link specific processes to specific anatomical subregions, but past studies of mPFC have yielded controversial results, leaving the precise nature of mPFC function unclear. To settle this debate, we recorded from the full dorso-ventral extent of mPFC in each of 8 rats, as they performed a complex economic decision task. These data revealed four distinct functional domains within mPFC that closely mirrored anatomically identified subregions, including novel evidence to divide prelimbic cortex into dorsal and ventral components. We found that dorsal aspects of mPFC (ACC, dPL) were more involved in processing information about active decisions, while ventral aspects (vPL, IL) were more engaged in motivational factors.

## Editor's evaluation

In this study, Diehl and Redish present a novel and fundamental account of functional variability in the rodent medial prefrontal cortex, in which the dorsal regions encode decision-related variables and the ventral regions encode variables linked to motivation. The study's experimental design is excellent, the analyses are appropriate, and the conclusions are based on compelling evidence. The suggestion of functional subdivisions in the prelimbic area is particularly provocative, and this conclusion, along with the data supporting it, will be of broad interest to those who study the anatomy and function of the rodent medial prefrontal cortex.

**\*For correspondence:**
redish@umn.edu

**Competing interest:** The authors declare that no competing interests exist.

## Introduction

The medial prefrontal cortex (mPFC) is considered central to the ability of mammals to succeed in the complex decision situations common to natural behavior (*Dalley et al., 2004*; *Euston et al., 2012*; *Miller and Cohen, 2001*). We engaged broad mPFC functioning using our unique Restaurant Row task: a foraging-based task that integrates sensory discrimination, value assessment, deliberative planning, re-evaluation of choice, and motivational factors (*Steiner and Redish, 2014*; *Sweis et al., 2018a*). Critically, Restaurant Row has been shown to engage mPFC at multiple stages of the task, with different causal manipulations of this brain area eliciting measurable changes in decision behavior (*Schmidt et al., 2019*; *Sweis et al., 2018b*; *Sweis et al., 2018c*). We coupled this strategy of engaging diverse mPFC functioning with neural ensemble recordings across the full dorso-ventral extent of mPFC. Using linear silicon probes that spanned nearly 3 mm of the medial wall, we simultaneously measured neural activity across the mPFC. This novel behavioral and physiologic approach to mPFC allowed us to identify the relationships between prefrontal activity and a wide array of variables relevant for task performance, to explicitly determine how mPFC firing patterns related to decision

processing at multiple well-controlled spatiotemporal moments in the task, and to directly compare these functional properties across the dorso-ventral anatomy of the medial prefrontal cortex.

The mPFC is typically described as consisting of the anterior cingulate cortex (ACC), the prelimbic cortex (PL), and the infralimbic cortex (IL), each of which have been identified with different functional roles (*Groenewegen and Uylings, 2000*; *Heidbreder and Groenewegen, 2003*; *Hoover and Vertes, 2007*; *Kolb, 1990*; *Sesack et al., 1989*; *Uylings et al., 2003*).

Previous studies have linked the anterior cingulate cortex (ACC) to three prominent functions, each associated with selection or evaluation of potential actions. One set of work has highlighted the role of ACC in choosing between a pair of options or general strategies, particularly within the context of shifting to a new strategy or diverting from a 'default' option (*Brockett et al., 2020*; *Ito et al., 2015*; *Karlsson et al., 2012*; *Lapish et al., 2008*; *Ma et al., 2016*; *Tervo et al., 2021*). Physiologic recordings of ACC have also been linked to evaluation of the outcome of a recent choice or of potential alternatives that were not taken (*Hyman et al., 2017*; *Hyman et al., 2013*; *Mashhoori et al., 2018*; *Sul et al., 2010*). Behavioral manipulations of ACC have similarly supported a role in the post-hoc assessment of decision making and use of that feedback to guide future choices (*Akam et al., 2021*; *Dalton et al., 2016*; *Hart et al., 2020*). Finally, a set of studies have identified a role for ACC in cost-benefit analyses, particularly when the cost entails exertion of physical or mental effort (*Hillman and Bilkey, 2010*; *Hillman and Bilkey, 2012*; *Hosking et al., 2014*; *Walton et al., 2003*).

In contrast, studies of the PL have pointed to a role in strategy representations, goal directed responses, and the expression of learned associations. This expression has been observed both for avoidance learning and fear conditioning (*Bravo-Rivera et al., 2014*; *Corcoran and Quirk, 2007*; *Diehl et al., 2018*; *Sierra-Mercado et al., 2011*; *Vidal-Gonzalez et al., 2006*). These learned associations have also been implicated within the context of goal directed decision making in which performance relies on acquired knowledge of a task structure (*Killcross and Coutureau, 2003*; *Tran-Tu-Yen et al., 2009*). Consistent with this idea, neural recordings from PL have revealed firing patterns associated with reward delivery, stimulus presentation, and specific behavioral response patterns (*Burgos-Robles et al., 2013*; *Hok et al., 2005*; *Passecker et al., 2019*; *Pratt and Mizumori, 2001*; *Takehara-Nishiuchi et al., 2020*). PL has also been implicated in the initial learning of these associations and in holding information online for a short period (*Burgos-Robles et al., 2009*; *da Silva et al., 2020*; *Gilmartin and McEchron, 2005*; *Gilmartin et al., 2013*; *Mukherjee and Caroni, 2018*; *Ragozzino et al., 1998*; *Seamans et al., 1995*), highlighting an alternative role in working memory. Finally, PL has been found to be important in shifting between strategies and the development of goal directed response patterns (*Birrell and Brown, 2000*; *Killcross and Coutureau, 2003*; *Powell and Redish, 2016*; *Ragozzino et al., 1999*; *Rich and Shapiro, 2007*; *Rivalan et al., 2011*; *Seamans et al., 1995*; *Tran-Tu-Yen et al., 2009*).

IL has been most studied within the contexts of fear conditioning and addiction. In particular, IL appears involved in extinction behavior and the shift toward new, learned associations that it entails (*Do-Monte et al., 2015*; *Milad and Quirk, 2002*; *Mukherjee and Caroni, 2018*; *Vidal-Gonzalez et al., 2006*). Interestingly, IL has been linked to extinction of both appetitive and aversive associations (*Bravo-Rivera et al., 2014*; *Capuzzo and Floresco, 2020*; *Laque et al., 2019*; *Milad and Quirk, 2002*; *Peters et al., 2008*; *Rhodes and Killcross, 2004*; *Riaz et al., 2019*; *Sierra-Mercado et al., 2011*), suggesting a more generalized role in suppression of past associations or replacement with newer ones. As it relates to decision making, IL has been implicated in the development and execution of habitual actions. Manipulations of the IL subregion prevent the development of goal-independent responding (*Coutureau and Killcross, 2003*; *Killcross and Coutureau, 2003*; *Smith and Graybiel, 2013*; *Smith et al., 2012*), and neural activity within IL is reminiscent of that in dorso-lateral striatum, a region associated with habitual action (*Smith and Graybiel, 2013*).

As will be described below, we present direct evidence that there are actually four functionally separable components of the mPFC (ACC, dorsal PL, ventral PL, IL), corresponding roughly to predictions from anatomical connectivity (*Berendse et al., 1992*; *Heidbreder and Groenewegen, 2003*; *Voorn et al., 2004*). Furthermore, these four components process information about different components of the Restaurant Row task, pointing to a coordinated engagement across mPFC subregions during complex decision making. We find the dorsal components more engaged in action selection components of the task, while the ventral components to be more engaged in motivational aspects of the task. Critically, these new understandings of mPFC function were possible because of our integrated

approach to the problem—engaging a wide array of functions via Restaurant Row, combined with simultaneously measuring neural activity across the full medial prefrontal cortex.

## Results

### Restaurant Row as an economic decision-making task

In order to effectively engage the mPFC during naturalistic decision making, we employed a complex foraging task that requires interaction across multiple different cognitive domains: Restaurant Row (*Steiner and Redish, 2014*; *Sweis et al., 2018a*). In the Restaurant Row task (RRow, *Figure 1A and B*), rats had 1 hr to forage for their day's food by making a series of decisions about whether or not to wait for delayed rewards at each of four food reward sites (restaurants), each providing a different flavor of food reward. At each restaurant, rats first entered the Offer Zone (OZ) where the frequency of an auditory cue signaled the temporal delay that the rat must wait to earn the food reward (1–30 s delays, drawn at random each visit). While the rat was in the OZ, the tone was played at the same frequency, once per second, until rats made a decision to either <u>accept</u> the offer by entering into the Wait Zone (WZ) where they would wait for their reward, or <u>skip</u> the opportunity, move to the Transition Zone (TZ), and advance to the next restaurant where they would receive a new offer. Upon accepting an offer and entering the WZ, the delay began to count down with tones played each second at progressively lower frequencies. The delay countdown/tones continued until either the delay reached zero and the respective reward was delivered (<u>earned</u>), or the rat elected to <u>quit</u> the offer and leave the WZ for the next site. Upon completing the delay and earning a reward, rats were denoted as being in the Reward Zone (RZ) during which time they could consume their reward and linger at the site until they elected to leave for the next restaurant. Note that while the OZ and TZ were spatially defined, the WZ and RZ were spatially coincident and were defined as before (WZ) and after (RZ) reward delivery respectively. In the event that rats rejected an offer, either by leaving the OZ in a skip or by leaving the WZ in a quit, the offer was rescinded and the tones ceased. Effective performance on RRow thus required assessment of the presented delay and the prospective reward that could be earned.

Consistent with previous studies (*Schmidt et al., 2019*; *Steiner and Redish, 2014*; *Sweis et al., 2018b*; *Sweis et al., 2018d*), we found rats' behavior during RRow to largely reflect the use of economically well-founded behavioral strategies (*Figure 1C–H*, *Figure 1—figure supplement 1*). Decisions in the task could be described by a threshold, with rats tending to accept and ultimately earn delay offers below threshold and skip or quit those above (*Figure 1C*). Importantly, these thresholds were derived from the subject's behavior and not imposed externally. Thresholds varied across rats and restaurant (*Figure 1D*) but were generally consistent from 1 day to the next for a given rat at a given restaurant (*Figure 1—figure supplement 1A, B*; Wilcoxon Signed-rank test of variance in thresholds across restaurants vs across sessions; rank = 2, p<0.05), indicating that they reflected subjective assessments of the worth of different reward flavors. In the OZ, accepting vs skipping an offer was closely tied to its distance from the threshold with progressively better offers accepted at higher rates and worse offers accepted at lower rates (*Figure 1E*). Likewise, in the WZ quit decisions were progressively more likely for poorer offers of longer delays that were farther above the threshold value (*Figure 1F*). These results suggest that we can consider the signed distance from threshold as a measure of subjective value (Value = threshold – offered cost). Note that in this equation, the threshold is a function of both rat and restaurant.

Decisions in the OZ were generally made quickly (1–3 s), with rats making accept decisions faster than skip decisions (*Figure 1—figure supplement 1C, D*; Wilcoxon Signed-Rank; z=–6.6, p<0.001). Among all accept choices, reaction time was inversely related to the subjective value of the presented offer with better offers leading to faster decisions (r=–0.30, p<0.001; n=8 rats and 20 value bins). This relationship however only held for accepted offers of positive value, or subjectively advantageous decisions (Positive Values: r=–0.31, p<0.01; Negative Values: r=0.08, p=0.46; n=8 rats and 10 value bins). The shorter reaction time for accept choices is consistent with earlier studies on the Restaurant Row task (*Sweis et al., 2018c*; *Sweis et al., 2018d*), and suggests that accepting an offer is the faster default response and skipping reflects a slower override process.

Quit decisions in the WZ were more common earlier in the delay period and average time to quit an offer was inversely related to its subjective value (*Figure 1—figure supplement 1E, F*; r=–0.52, p<0.001). In the RZ, rats generally lingered at least 5 s after earning their rewards with a skewed

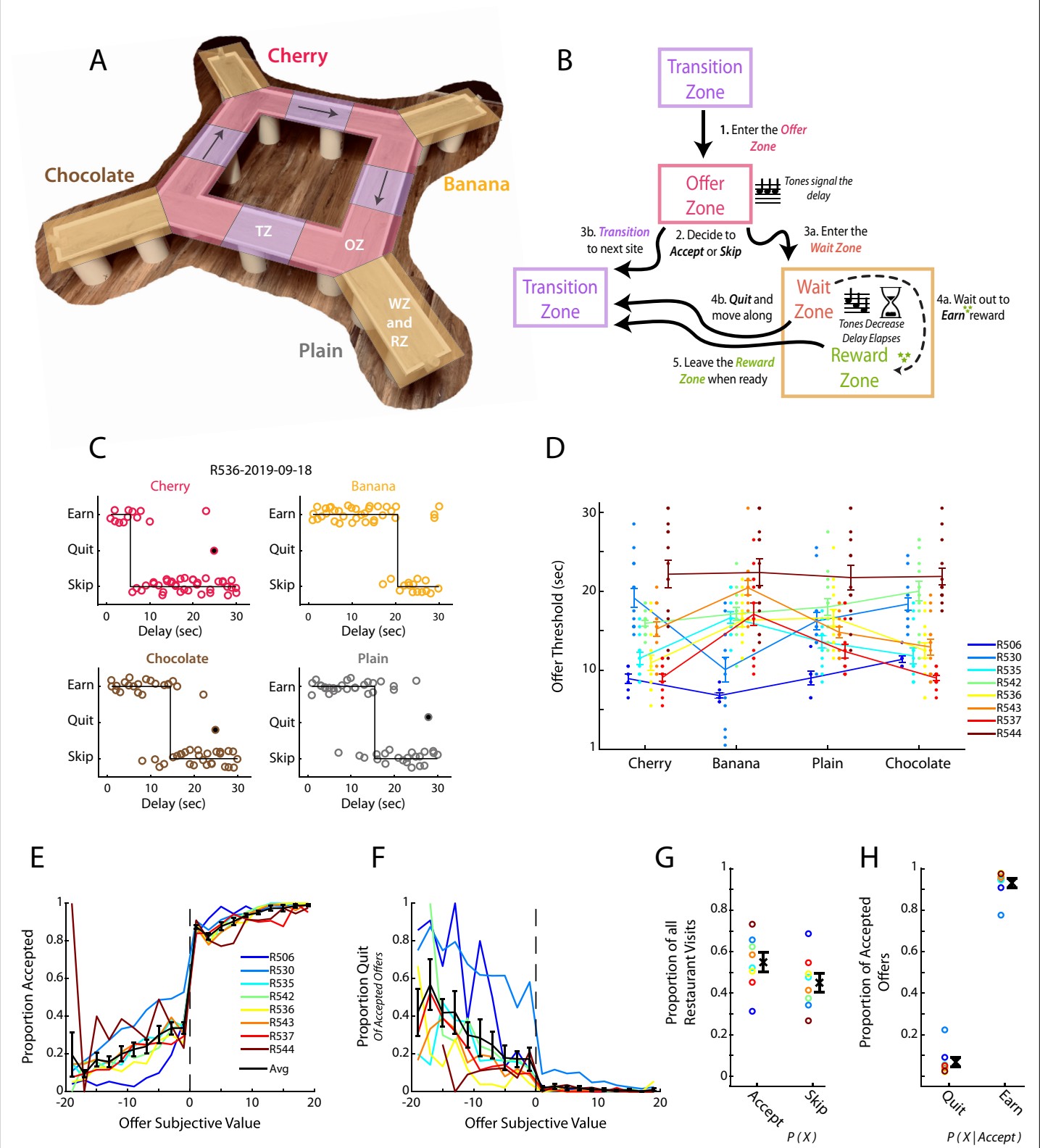

**Figure 1.** Restaurant Row as a decision-making task. (**A**) Illustration of the Restaurant Row (RRow) maze. (**B**) Schematic of the task structure and the decision progression through an individual reward site (restaurant). (**C**) Example choice responding and thresholds across one behavioral session on RRow. Each circle represents one restaurant visit and is plotted according to the presented delay and the behavioral response. Offers that were quit are shown as filled circles. (**D**) Thresholds across the four restaurants for each rat in each session (n=8 Rats; 104 Total session). Threshold Means ± SEM

*Figure 1 continued on next page*

*Figure 1 continued*

for each rat are shown as error bars. (**E, F**) Proportion of offers that were accepted (**E**) or Quit (**F**) as a function of the subjective value of the offer (Value = Threshold – Offer Delay). Data are presented for each rat with the Mean ± SEM across rats shown in black. Note that Quits (**F**) are calculated as the proportion of offers that were initially accepted. (**G**) Proportion of all restaurant visits that were Accepted or Skipped. (**H**) Proportion of Accepted offers that were subsequently Quit or Earned.

The online version of this article includes the following video and figure supplement(s) for figure 1:

**Figure supplement 1.** Additional behavior on Row.

**Figure supplement 2.** Localization of single-unit recordings from medial prefrontal cortex.

**Figure supplement 3.** Basic spiking characteristics of medial prefrontal cells.

**Figure 1—video 1.** 3D Anatomical location of recorded mPFC cells.

https://elifesciences.org/articles/82833/figures#fig1video1

---

distribution including long, relatively infrequent linger bouts (*Figure 1—figure supplement 1G*). Interestingly, we did not find a significant relationship between linger time and offer value (*Figure 1—figure supplement 1H*; $r=0.05$, $p=0.51$) in contrast to previous studies (*Sweis et al., 2018a*).

## Firing of prefrontal cells is related to ongoing behavior

To record large ensembles of precisely localized mPFC units we implanted rats with linear, 64 channel silicon probes directed along the full extent of the medial wall of the prefrontal cortex (n=13 total probes; *Figure 1—figure supplement 2A*, *Table 1*). All recordings were taken after rats were well trained on Restaurant Row, highly accustomed to performing the task under recording procedures, and exhibiting stable behavioral performance. Post-hoc histological verification of probe locations confirmed that 11 of the 13 probes were completely contained within the mPFC (ACC, PL, or IL regions). The remaining two probes were in the VO and MO regions and these data were excluded from all analyses. In total, we recorded 3017 single units from the mPFC of eight rats (104 total sessions; *Table 2*) and precisely localized each one within a standardized three-dimensional space (*Figure 1—video 1*; *Figure 1—figure supplement 2D*) using a combination of waveform profiles across recording channels (*Csicsvari et al., 2003*), histological information, and reconstruction of probe movement through the brain (*Figure 1—figure supplement 2*; see Methods). Of these 3017 recorded cells, we identified putative principal cells (n=2441) and interneurons (n=576) on the basis of cells' average firing rates and the width of their spike waveform (*Figure 1—figure supplement 3A, B*).

As a first assessment of neural activity in mPFC, we computed a variety of basic spiking and waveform characteristics (*Figure 1—figure supplement 3C*). To then directly evaluate if these basic physiologic properties varied systematically across the mPFC, we arranged cells according to their location along the DV axis. Overall, we found a broad distribution of basic spiking properties and, outside of the division between principal cells and interneurons, no clear groupings emerged. However, a few basic properties varied systematically along the dorso-ventral axis. Overall firing rate was higher in

---

**Table 1.** Probe counts by rat.
Breakdown of the sex and allocation of probe targeting across rats.

| Rat ID | Sex | Number of Probes | Dorsal Probe (ACC/PL) Hemisphere | Ventral Probe (PL/IL) Hemisphere |
|--------|-----|------------------|-----------------------------------|-----------------------------------|
| R506 | Male | 1 | Right | *N/A* |
| R530 | Female | 2 | Left | *Right (In MO; Excluded)* |
| R535 | Female | 2 | Left | *Right (In VO; Excluded)* |
| R542 | Male | 2 | Right | Left |
| R536 | Female | 1 | Left* *Single probe moved from dorsal to ventral* | |
| R543 | Male | 1 | *N/A* | Right |
| R537 | Female | 2 | Right | Left |
| R544 | Male | 2 | Right | Left |

Table 2. Cell counts by rat.

Breakdown of the number of cells recorded from each rat. Blue shaded cells denote the sum across a respective column and green shaded cells denote the mean. Note that while a large proportion of ACC and dPL cells come from one rat each, for all analyses we confirmed that results were qualitatively similar across the remaining animals (data not shown).

| Rat ID | Total Sessions | Total Cells | Per Session Counts | | | | Totals Counts | | | | Per Session Averages | | | |
|---|---|---|---|---|---|---|---|---|---|---|---|---|---|---|
| | | | Avg Cells | Min Cells | Max Cells | Std Cells | ACC Cells | dPL Cells | vPL Cells | IL Cells | ACC Cells | dPL Cells | vPL Cells | IL Cells |
| R506 | 7 | 109 | 15.6 | 7 | 20 | 4.6 | 41 | 68 | 0 | 0 | 5.9 | 9.7 | 0.0 | 0.0 |
| R530 | 14 | 206 | 14.7 | 3 | 34 | 10.6 | 0 | 92 | 102 | 12 | 0.0 | 6.6 | 7.3 | 0.9 |
| R535 | 13 | 611 | 47.0 | 35 | 63 | 9.9 | 0 | 563 | 48 | 0 | 0.0 | 43.3 | 3.7 | 0.0 |
| R542 | 14 | 562 | 40.1 | 29 | 50 | 5.7 | 1 | 40 | 310 | 211 | 0.1 | 2.9 | 22.1 | 15.1 |
| R536 | 14 | 145 | 10.4 | 3 | 24 | 5.8 | 12 | 44 | 46 | 43 | 0.9 | 3.1 | 3.3 | 3.1 |
| R543 | 14 | 464 | 33.1 | 19 | 47 | 8.0 | 0 | 9 | 172 | 283 | 0.0 | 0.6 | 12.3 | 20.2 |
| R537 | 14 | 530 | 37.9 | 20 | 54 | 10.8 | 14 | 2 | 157 | 357 | 1.0 | 0.1 | 11.2 | 25.5 |
| R544 | 14 | 390 | 27.9 | 18 | 41 | 7.1 | 4 | 73 | 177 | 136 | 0.3 | 5.2 | 12.6 | 9.7 |
| TOTAL | 104 | 3017 | 28.3 | 16.8 | 41.6 | 7.8 | 72 | 891 | 1012 | 1042 | 1.0 | 8.9 | 9.1 | 9.3 |

dorsal aspects of mPFC, whereas spiking was more variable in ventral aspects. Though, in all cases there remained large amounts of overlap in the full distribution of mPFC spiking characteristics. Thus, overall basic spiking characteristics throughout the mPFC were highly diverse, with some properties exhibiting significant, although limited, gradation along the dorso-ventral axis.

To evaluate how mPFC activity was related to the behavioral execution of the RRow task, we directly compared the firing of mPFC cells with a series of task relevant behavioral variables. We observed many individual cells with clear relationships between their firing rate and relevant task variables including: the accept/skip decision made in the Offer Zone, the temporal delay of the presented offer, the overall time that had elapsed through the behavioral session, and the subjective rank of the prospective reward (*Figure 2A*). Importantly, all the task relevant variables that we analyzed were significantly represented within the mPFC population (*Figure 2B and C*; Wilcoxon Signed-Rank vs Shuffled Spikes; all $z > 23.5$, all $p < 0.001$). Notably, the proportion of cells with significant relation to behavior was not equivalent across all behavioral variables. Furthermore, representations of task variables were highly distributed, with cells often exhibiting a significant correlation to many different variables (*Figure 2D and F*). Importantly, while some behavioral variables in RRow were strongly correlated (e.g. OZ choice depends on offer delay and the rat's subjective threshold; *Figure 2E*), multimodal coding by mPFC cells was clear and obvious even across completely unrelated behavioral dimensions. As an additional control, we performed a pair of stepwise regression procedures to account for correlations between behavioral variables (*Figure 2—figure supplement 1*). Even when progressively removing the influence of each behavioral variable on mPFC firing, we still observed multimodal coding of behavioral variables across the population. These findings highlight that mPFC is strongly engaged during RRow with neural activity patterns that are relevant to the decision processes of this task, and are consistent with other studies of mPFC function that find mixed selectivity tuning in prefrontal cells (*Duncan, 2001*; *Powell and Redish, 2014*; *Rigotti et al., 2013*; *Seamans et al., 2008*; *Stokes et al., 2013*; *Wallis et al., 2001*).

## Four distinct functional subdivisions exist within the mPFC

To directly examine how information is processed across the mPFC, we computed the functional coupling between pairs of simultaneously recorded cells using transfer entropy (TE). TE is a measure between a pair of cells that quantifies the degree to which knowing about one cell's activity allows one to describe that of another cell's activity, above and beyond what can be known from the second cell's own past activity (*Timme and Lapish, 2018*). It therefore provides a metric of the functional communication between pairs of cells and can provide insight into information flow through a network. Furthermore, we leveraged the ability provided by silicon probes to precisely localize recorded cells in the brain in order to specifically examine the relay of information across the prefrontal dorso-ventral axis.

While the firing of some mPFC cell pairs was unrelated (23%), overall, the distribution of TE was significantly above chance level (Wilcoxon Signed-Rank, $z = 230.1$, $p < 0.001$) with a strong rightward skew, indicating substantive communication between select pairs of cells (*Figure 3A*). Organizing cell pairs according to their respective anatomical locations provided direct insight into the nature of these high-coupling relationships (*Figure 3B*). TE was elevated along the identity diagonal, indicating increased communication between cells that were located at similar positions. Yet importantly, this increase was not uniform across the full dorso-ventral extent, but rather was characterized by distinct anatomical positions in which TE was elevated. In other words, functional communication was enriched at a handful of specific locations within the mPFC, indicative of localized processing subdivisions. We quantitatively identified these subdivisions, and the boundaries between them, by averaging the TE data from bins proximal to the identity diagonal (within 200 μm) at each dorso-ventral depth along the medial wall (*Figure 3C*). This analysis revealed four peaks separated by identifiable valleys between them (data were not unimodal; Hartigan's Dip Test, $D = 0.0291$, $p < 0.001$). Thus, examination of communication between simultaneously recorded cells revealed that during this decision-making task, the mPFC was characterized by four distinct functional subunits.

While TE calculated throughout the full behavioral period revealed distinct subunits within mPFC, RRow is an inherently dynamic task with distinct phases of choice, re-evaluation, reward, and transition (*Steiner and Redish, 2014*; *Sweis et al., 2018a*; *Sweis et al., 2018b*; *Sweis et al., 2018c*; *Sweis et al., 2018d*). Thus, we repeated our TE analysis, taking spiking data from each of the four task phases independently (*Figure 3—figure supplement 1*). Notably, data from each task phase

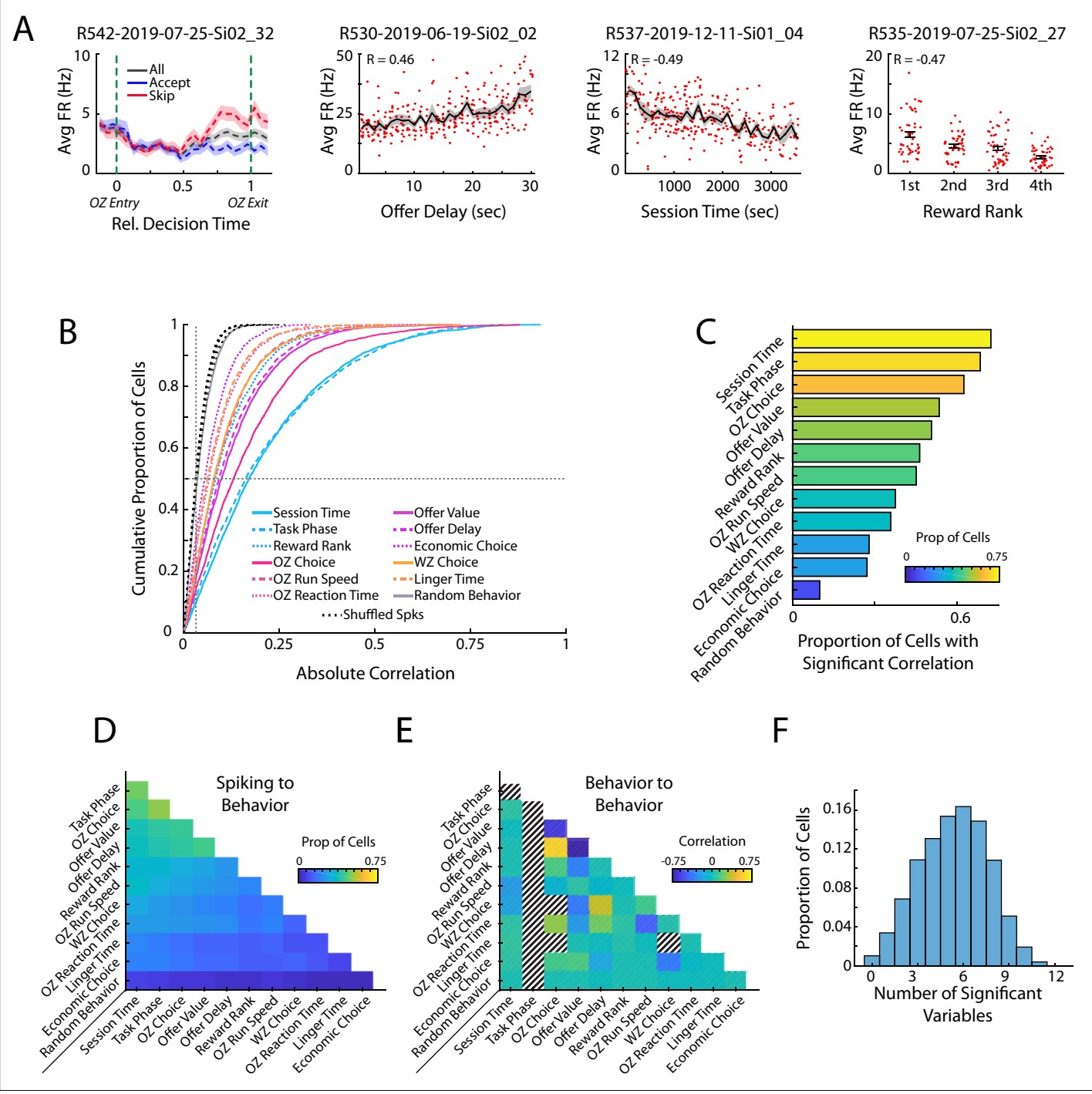

**Figure 2.** Prefrontal cells respond to task relevant variables. (**A**) Four example mPFC cells that respond to offer zone choice, offer delay, session time, and reward rank. Mean and standard errors are shown. (**B**) Cumulative density functions of the unsigned correlation between firing rate of mPFC cells (n=3017) and various task relevant behaviors. Chance values are computed by correlating firing rates to randomly drawn values (Random Behavior) and by time shifting spiking of each cell and correlating the dissociated spiking data to the behaviors of interest (Shuffled Spks). Firing rates in mPFC were significantly more correlated to each behavioral variable than to the respective chance distributions. (**D**) Proportion of the mPFC cell population that was significantly correlated to both of a pair of behavioral variables. (**E**) Different pairs of behavioral variables were correlated to each other. The average correlation across the 104 behavioral sessions is shown for each pair of variables. Note that Task Phase varied within each restaurant visit and thus could not be compared to other variables. Additionally, some combinations of variables were conditionally impossible to co-occur and thus could not be correlated (e.g. OZ Choice and WZ Choice). (**F**) Individual mPFC cells were often significantly correlated to multiple behaviors. Distribution of

*Figure 2 continued on next page*

*Figure 2 continued*

multimodal coding across the mPFC cell population. Each bar indicates the proportion of mPFC cells with firing rates that were significantly correlated to the indicated number of behavioral variables.

The online version of this article includes the following figure supplement(s) for figure 2:

**Figure supplement 1.** Stepwise regression of spiking to behavior.

**Figure supplement 2.** Segregation of prefrontal responses by subregion.

produced comparable TE profiles with comparable subregional boundaries. These similarities indicate that the functional subregions are consistent throughout the RRow task structure and strongly suggest that they reflect a generalized processing architecture within mPFC.

Importantly, these subdivisions that were identified purely functionally largely matched the stereo-taxically defined subdivisions identified in previous anatomical studies, as outlined in the Paxinos and Watson atlas (*Figure 3D*; *Paxinos and Watson, 2007*). Functionally defined boundaries at the dorsal and ventral end of the mPFC closely aligned with the anatomically defined boundaries between ACC and PL dorsally and between PL and IL ventrally. Accordingly, we will designate our most dorsal and ventral functional subunits as ACC and IL respectively. However, our functional analysis also revealed a transition in the intermediate range of the mPFC, roughly bisecting the anatomical PL subregion. This finding implies that PL is not a functionally homogenous area, but rather should be further split into dorsal (dPL) and ventral (vPL) subregions, a premise that has some historical anatomical support (*Berendse et al., 1992*; *Heidbreder and Groenewegen, 2003*; *Vogt and Paxinos, 2014*; *Voorn et al., 2004*). While this anatomical separation has not yet been widely adopted, particularly in functional studies, our data strongly suggest that these regions need to be considered separately.

Having identified the existence of four distinct functional subregions within mPFC (ACC, dPL, vPL, IL), we examined the degree of information transfer between these four areas during the full behavioral session (*Figure 3F*). In line with the fact that subregions were originally identified as areas with elevated communication, TE between cell pairs was largest when both cells were located within the same subregion (intra-regional communication). Functional communication between cell pairs that spanned different subregions (inter-regional communication) was significantly weaker (Mann-Whitney U test, Intra vs Inter region; z=19.87, p<0.001). Interestingly, functional connectivity varied as a function of distance between subregions, with adjacent areas exhibiting stronger interactions than more distally located areas (ANOVA grouped by Subregion Step; F(3) = 179.5, p<0.001; HSD post-hoc: Step 0 vs 1, p<0.001, Step 1 vs 2, p<0.001, Step 2 vs 3, p=0.76; correlation between pairwise TE and subregion steps: r=–0.07, p<0.001). We also examined the neural representation of our array of task relevant variables in each subregion independently (*Figure 2—figure supplement 2*). Importantly, we found each behavioral variable to be represented across all four areas, although the overall degree of this coding varied between subregions, suggesting differential processing of task relevant information between them. Finally, because our data set contained comparatively fewer ACC cells relative to the other subregions, we performed a bootstrapping procedure in which we repeated our analyses after down-sampling the data from the dPL, vPL, and IL subregions to match that of ACC. Importantly, even with a subsampled data set, we identified subregional boundaries and strength of inter-regional communication levels that matched our main findings (*Figure 3—figure supplement 2*).

## Generalized task variables are differentially represented across mPFC subregions

Although TE measures revealed the existence of four distinct functional subregions, it only measures the extent of information transfer, and does not provide any insight into what information is represented within these brain areas as decisions are made. Mirroring natural conditions, decision making in the RRow task is not performed as isolated events but rather in a broader task context: The specific nature of decision that must be made varies according to the current behavioral phase in the task (task phase). Rats are forced to operate under a global time constraint both achieving satiety and reducing their overall opportunity over the course of the session (session time). Economic decisions in the Offer Zone are made according to the subjective quality of the currently available reward (reward rank) and the offer presented to the rat (offer delay). Importantly, each of these broader task components were encoded across all four mPFC subdivisions, although to varying degrees (*Figure 4A*; Wilcoxon

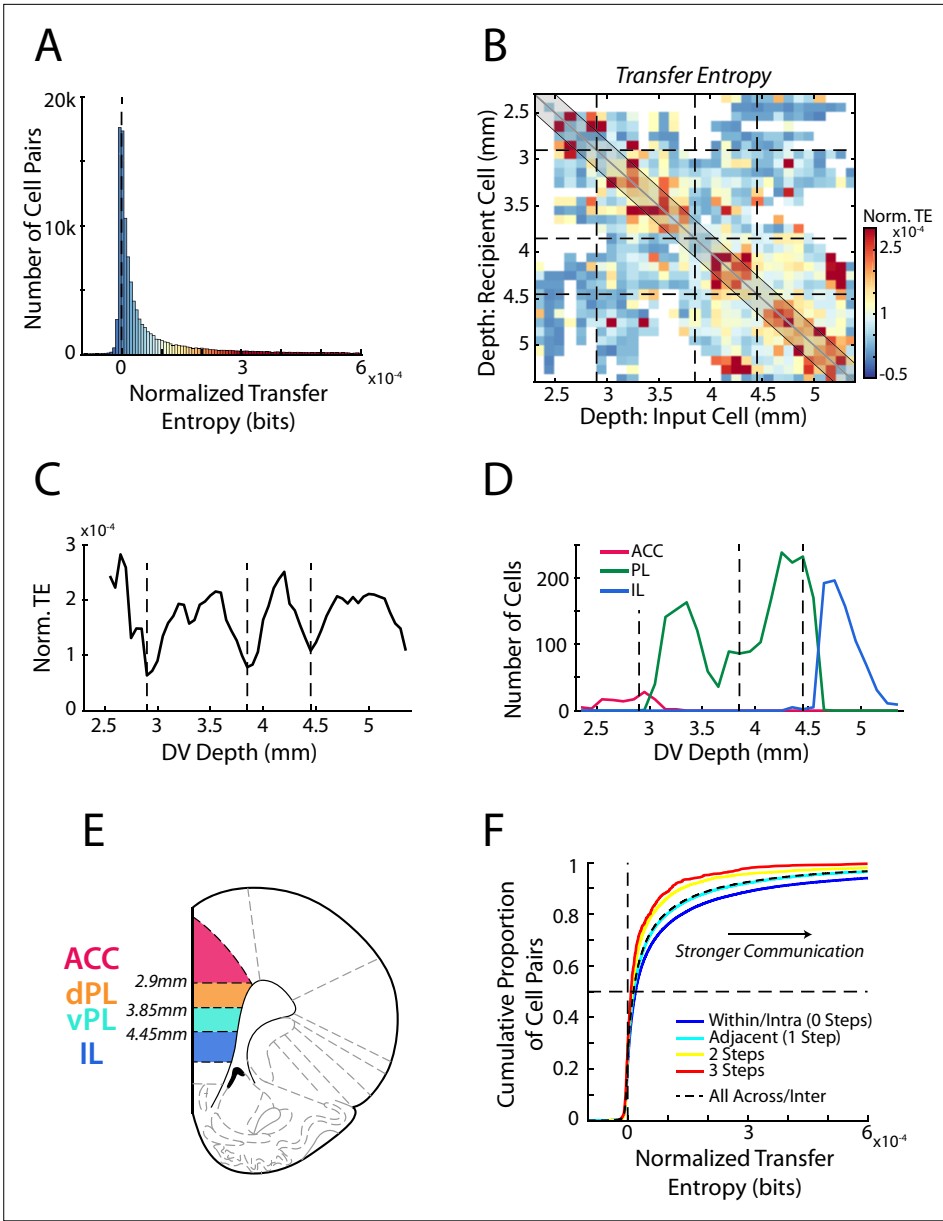

**Figure 3.** Transfer entropy reveals subregions within mPFC. (**A**) Transfer entropy (TE) between simultaneously recorded pairs of cells (n=107,418 pairs) while rats performed RRow. Data were normalized based on shuffled spike times of the input cell meaning that TE values above zero (dashed line) reflect cell interactions that are stronger than expected by chance. Across the population, TE was significantly higher than chance. Data along the x-axis are color coded according to the scale in panel B. (**B**) Mean normalized TE (norm. TE) between pairs of cells as a function of the DV location of the input and recipient cells. Black dashed lines represent the functional boundaries between mPFC subregions computed in panel C. (**C**) Quantification of TE between cell pairs within 200 μm of the identity diagonal (shaded region in B). Data are the mean TE of bins within 200 μm of each DV depth. Dashed lines denote the local minima of the curve and represent derived transition bounds between subregions. (**D**) Number of recorded ACC, PL, and IL cells at each DV position, as classified based purely on anatomical information as defined in the Paxinos and Watson atlas. TE derived bounds are repeated from panel C. (**E**) A schematic of four subdivisions of mPFC based on the boundaries derived from analysis of functional communication (TE). (**F**) Cumulative distributions of normalized TE values between pairs of cells. Data are grouped according to the relative proximity (number of steps) between the subregions of the two cells. Black dashed line groups all inter-regional pairs regardless of distance. Note that a more rightward distribution corresponds to overall stronger TE.

The online version of this article includes the following figure supplement(s) for figure 3:

**Figure supplement 1.** Transfer entropy analyses during different task phases.

*Figure 3 continued on next page*

*Figure 3 continued*
**Figure supplement 2.** Subsampling of mPFC recordings for TE calculations.

Signed-Rank: Task Phase, z=47.5, p<0.001; Session Time, z=45.4, p<0.001; Reward Rank, z=41.9, p<0.001; Offer Delay, z=35.5, p<0.001; OZ Choice, z=44.0, p<0.001; ANOVA across variables; F(4) = 925.1, p<0.001). For each variable, we evaluated the degree of neural coding by calculating the mutual information between the firing rates of mPFC cells and the behavior of interest. Briefly, mutual information quantifies the degree of information about a particular behavior of interest that can be extracted from neural spike rates and describes how well neural spiking can differentiate between behaviors (*Timme and Lapish, 2018*).

Success in the RRow task requires an understanding of the current phase of the task and its objectives, and an ability to employ an appropriate behavioral strategy given that understanding. In line with a role of the mPFC in executive functioning, we found strong coding for task phase (OZ, WZ, RZ, or TZ) in each of the four prefrontal subregions (Wilcoxon Signed-Rank: all z>7.4, all p<0.001). However, mutual information between cell firing rate and task phase was not equivalent across subregions (*Figure 4B and C*; ANOVA across subregions; F(3) = 39.3, p<0.001). Instead, this information was most strongly represented in ACC and dPL, weaker in vPL, and then elevated again in the ventral-most IL subregion (ANOVA HSD post-hoc: all pairwise comparisons p<0.001). In general, average neuronal firing rates were highest while rats were in the OZ making their decisions, and then tended to decrease during the WZ and the RZ. These variations in firing rate across task phase were strongest in the ACC subregion, as would be expected based on mutual information calculations.

At the broader task level, RRow also requires an understanding of the progression through the behavioral session. Because the task design limits rats to only 1 hr to earn rewards, progression through the session is inversely related to the overall opportunity remaining. In parallel, progression through the session is intertwined with other variables likely to influence value-based decisions such as satiety, fatigue, and general motivation. Again, we found significant coding for elapsed time into the session across all four mPFC subregion (Wilcoxon Signed-Rank: all z>5.9, all p<0.001). However, in contrast to task phase, neural coding for session time was strongest in the ventral components of mPFC, vPL and IL (*Figure 4D and E*; ANOVA across subregions; F(3) = 21.4, p<0.001; HSD post-hoc: ACC vs dPL, p=0.24; vPL vs IL, p=0.66; all other comparisons p<0.001). Interestingly, on average the firing rates of cells in vPL and IL were strongly elevated in the first third of the session and then decreased and remained stable for the remaining duration. In contrast, firing rates in ACC were initially low at the beginning of the session, and then increased over the initial 10 min. In light of these changes in mPFC activity over the course of the session, we revisited our main analyses of mPFC functioning across subregions to determine the degree to which they varied as motivational factors evolved over the session (*Figure 4—figure supplement 1*).

Lastly, RRow decision making involves evaluating a presented offer within the context of the subjective value of a prospective reward option. Because the location of each reward flavor remained static throughout the entire behavioral sequence, rats could easily predict the potential reward at each restaurant visit and appropriately evaluate its subjective worth. Neural coding for this information about prospective rewards was present in each mPFC subregion (*Figure 4F and G*; Wilcoxon Signed-Rank: all z>7.0, p<0.001). Yet across the four subregions mutual information about reward ranking was most robust in the dorsal areas (ACC and dPL) and weaker in the ventral areas (vPL and IL) (ANOVA across subregions; F(3) = 45.4, p<0.001; HSD post-hoc: ACC vs dPL, p=0.29; vPL vs IL, p<0.05; all other comparisons p<0.001). Interestingly, examining raw firing rate differences across reward ranks revealed that in ACC and dPL, firing rates tended to be lower for the more preferred reward options.

In conjunction with assessing the subjective quality of their potential reward, rats must evaluate the delay of the offer presented to them in making their choice. We observed significant coding of offer delay throughout the mPFC (*Figure 4H*; Wilcoxon Signed-Rank: all z>6.9, all p<0.001). Yet by nature, accept/skip decisions on RRow should be inexorably linked to the offer delay, with rats that behave economically largely accepting short delays and skipping long delays (*Figures 1E and 2E*). Accordingly, we also found strong coding of offer zone choice across the mPFC (*Figure 4H*; Wilcoxon Signed-Rank: all z>7.3, all p<0.001) which was strongest in the dorsal subregions (ANOVA across subregions; F(3) = 134.0, p<0.001; HSD post-hoc: vPL vs IL, p<0.05; all other comparisons p<0.001). To distinguish which of these related variables was more closely tied to mPFC firing we split our data

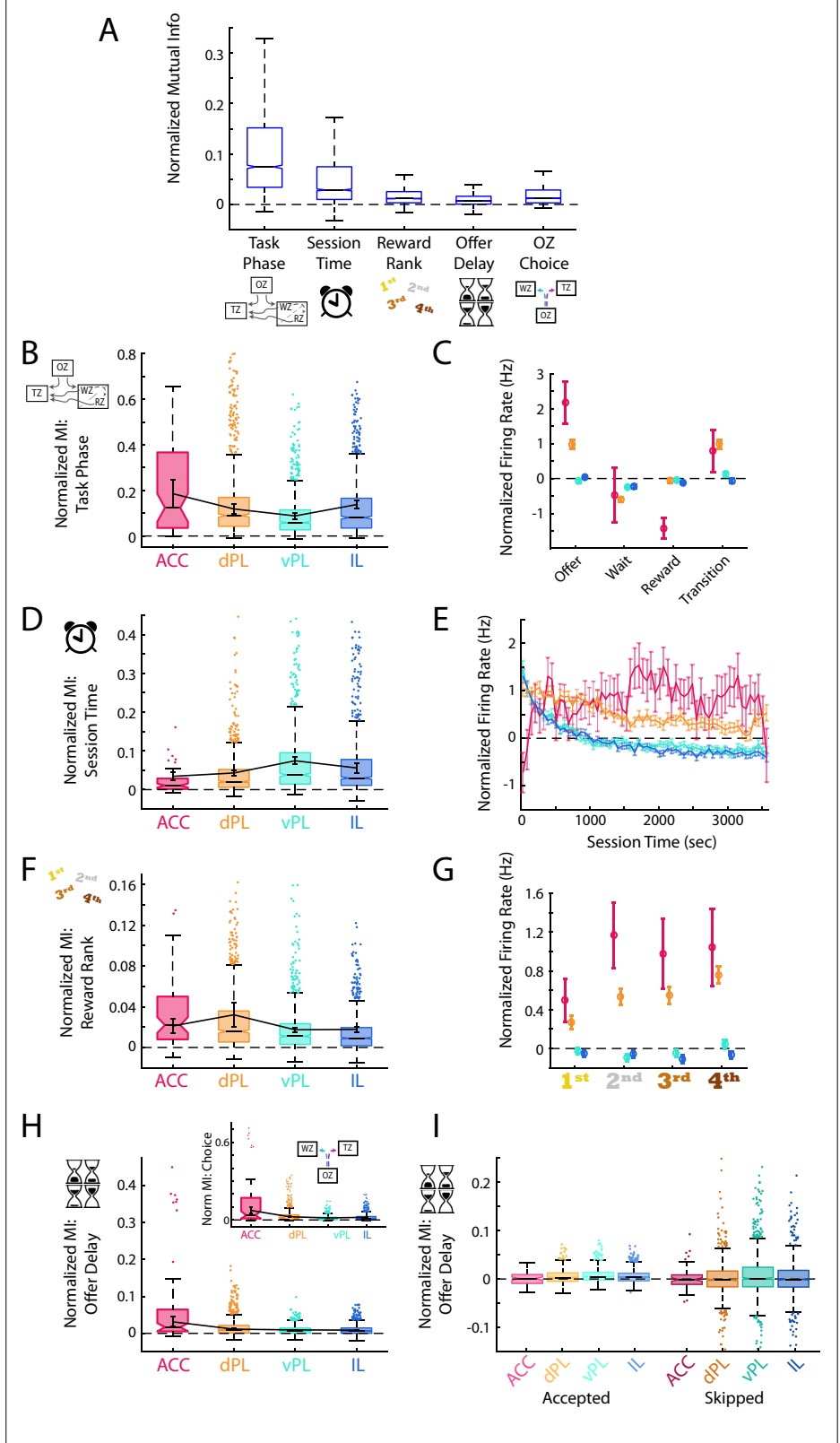

**Figure 4.** Generalized task variables are differentially represented. (**A**) Mutual information (MI) calculations for mPFC cells between the task phase, elapsed session time, reward site ranking, offer delay, and choice made in the offer zone. For each cell, MI values were normalized based on data generated with shuffled spike times. For clarity, distribution outliers (beyond 1.5 x IQR) are not displayed. (**B, D, F, H**) Normalized MI to task phase (**B**), session

*Figure 4 continued on next page*

*Figure 4 continued*

time (**D**), reward rank (**F**), and offer delay (**H**) for each mPFC subregion. Boxplots with outliers show the data for all recorded cells in each subregion. Data were also averaged within each rat and the Mean ± SEM across rats is shown overlayed in black. The inset in panel (**H**) shows the corresponding MI data for offer zone choice. Note that because choice is based largely on the presented delay these two data are highly correlated. (**C, E, G**) Mean ± SEM firing rate of cells in each mPFC subregion across the four task phases (**C**), over the session duration (**E**), and for each restaurant rank (**G**). Note that firing rates were normalized based on shuffled spike times. Reward rank was determined in each session by sorting restaurants by decision threshold with the largest threshold (willing to wait longest to earn a reward) ranked 1$^{st}$. (**I**) MI for offer delay split by whether rats accepted or skipped in the OZ. Note that after accounting for the choice made, MI for delay is largely eliminated suggesting that the neural responses are more associated with choice than delay.

The online version of this article includes the following figure supplement(s) for figure 4:

**Figure supplement 1.** Analysis of main functional effects over the session.

according to OZ choice (Accept or Skip) and recalculated the MI between firing rate and offer delay (*Figure 4I*). Interestingly, coding for offer delay was largely eliminated when accounting for OZ choice (Accept: ACC z=–0.2, p=0.79; dPL, vPL, IL all z>9.0, all p<0.001; Skip: ACC, dPL, IL all Z<0.5, all p>0.05; vPL z=3.4, p<0.001), suggesting that neural responses were more tightly associated with rats' decision making in the Offer Zone than the offer delay that had been presented to them.

Integrating our findings across these behavioral variables implies a biasing of functional information across the mPFC. Representations of task phase, reward ranking, and OZ choice were stronger in dorsal portions (ACC and dPL) of mPFC, while session time was more strongly represented in ventral portions of mPFC (vPL and IL). Both task phase and reward ranking are important components of successful decision-making processing in Restaurant Row. Decisions in the OZ, WZ, RZ, and TZ all encompass different considerations that must be accounted for when making decisions, but reward preferences play a role in each. Our data suggest a preferential involvement of dorsal mPFC in these processes. In contrast, the increased representations of progress through the session in ventral portions of mPFC (vPL and IL) suggest that more generalized and prolonged motivational aspects preferentially engage the ventral components of mPFC.

## Activity in mPFC strongly encodes the identity of the upcoming choice

In the RRow task, value-based decisions occur predominantly within the Offer Zone (*Figure 1G and H*). Therefore, examination of the firing rate profiles of mPFC cells while rats are in the OZ can provide insight into how neural processing differs when rats accept a presented offer vs when they elect to skip the offer and transition to the next restaurant. We directly measured these differences in neural activity by computing the mutual information between cell firing rates over the decision period (time normalized across visits to equate durations) and the choice that was ultimately made. Across all four subregions, coding for the eventual choice was initially at chance level as rats approached the OZ, prior to receiving information about the delay to earn reward (*Figure 5A*). However, shortly after entering the OZ, coding for the eventual choice increased above chance and remained elevated through the end of the decision period (Repeated measures ANOVA for subregions over time bins; Interaction: F(75) = 87.9, p<0.001. Post-hoc on each time bin via one sided Wilcoxon Signed-rank vs 0 corrected for multiple comparisons). Importantly, this choice coding emerged strongest within the ACC population where mutual information for choice peaked just prior to rats exiting the OZ and registering their choice, although it was also quite prominent in dPL and was detectable in vPL and IL (ANOVA grouped by subregion; F(3) = 144.4, p<0.001; HSD Post-hoc: ACC vs others, all p<0.001). Directly examining the net firing rate of mPFC cells between accept decisions and skip decisions pointed to important differences across the subregions (*Figure 5B*). Net changes in FR were dramatically larger in ACC and dPL as compared to ventral mPFC (vPL and IL), a result consistent with the differences in magnitude of mutual information. Interestingly, net firing rates also revealed that ACC cells tended to be more responsive during skip decisions (68% net skip; $X^2$ test for equal proportions vs 50%, $X^2$=4.9, p<0.05), whereas dPL cell activity was more biased toward accept decisions (55% net accept; $X^2$=4.8, p<0.05). In the ventral areas, vPL was again biased toward skip decisions (55% net skip; $X^2$=6.2, p<0.05), but the preferred responses across the IL population was balanced between accept and skip choices (50% net skip; $X^2$=0.0, p=0.93). These features of net firing rate highlight

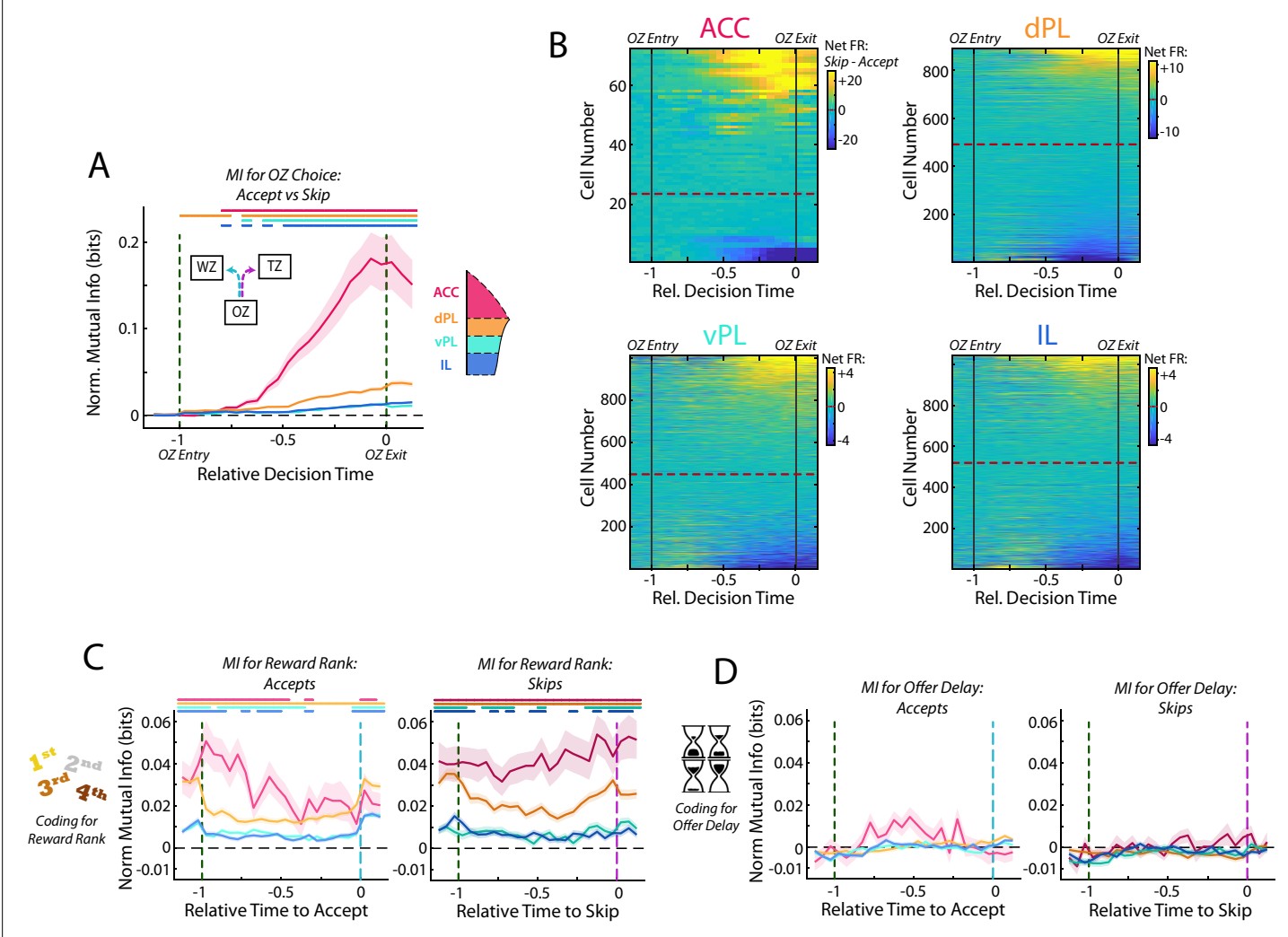

**Figure 5.** Activity in mPFC strongly encodes the identity of the upcoming choice. (**A**) Mutual information (MI) between firing rate and OZ choice (accept vs skip) for cells in each mPFC subregion. Note that durations in the OZ are normalized for each lap to yield relative duration for analysis. (**B**) Net firing rate of mPFC cells between accept and skip decisions (Skip FR – Accept FR). For each subregion, cells are sorted according to the net firing rate after entering the OZ. Color scales are identified for each subregion but note the change in scale between subregions. (**C**) Mutual information between firing rate of mPFC cells and restaurant ranking and (**D**) between firing rate and the presented offer delay. Data are segregated according to the decision that was made. For (**A,C,D**), data are shown as the Mean ± SEM across cells in each mPFC subregion. All data were normalized against analyses performed using shuffled spike times. Time bins with a significant MI response (one-tailed Wilcoxon Signed-rank, α=0.05 after multiple comparison correction; n = 26 time bins) are identified by colored bars above each plot.

The online version of this article includes the following figure supplement(s) for figure 5:

**Figure supplement 1.** Offer Zone responses by recording hemisphere.

**Figure supplement 2.** Multiplexed coding of Rank and Delay.

**Figure supplement 3.** Responses in the Offer Zone from subsampled data.

important differences between prefrontal subregions, and in the case of ACC fits with a body of literature pointing toward a role in enabling the agent to abandon an available reward in favor of other options (skips in this task; *Hayden et al., 2011*; *MacDonald et al., 2000*; *Shenhav et al., 2013*; *Tervo et al., 2021*).

In the RRow task, rats register their decision, accept or skip, via a motor action either into the Wait Zone or the Transition Zone. Thus, cognitive decisions and motor actions are inherently intertwined. In order to dissociate between decision making and motor execution, we leveraged the fact that accept decisions in RRow always involve a left turn whereas skip decisions are always a right turn.

Accordingly, neurons in the left hemisphere will be ipsilateral to the motor output of an accept action and contralateral to that of a skip. Past work in the cingulate cortex and related premotor structures suggests that directed motor responses preferentially engage contralateral neural systems (*Erlich et al., 2011*). Therefore, if our neural responses were tied to motor actions, we would expect that left hemisphere cells should be preferentially engaged during skip decisions (contralateral actions), whereas right hemisphere cells should be engaged during accept decisions. To test this, we segregated our recordings according to hemisphere (*Figure 5—figure supplement 1*). Critically, we found comparable neural responses from both left and right hemispheres with no strong bias toward preferential firing toward a contralateral motor action. This then suggests that our observed neural changes throughout mPFC are more likely related to decision making than to motor outputs.

Optimal decision making on Restaurant Row requires that rats determine the subjective value of a presented offer by integrating information about identity of the current restaurant and the offer delay. Thus, we examined the degree to which current restaurant and offer delay were individually represented within the mPFC. Consistent with our prior analysis across the full task period, coding for the identity of the current restaurant was abundant across all four mPFC subregions while rats were in the OZ, although they differed in overall magnitude (*Figure 5C*; Rm ANOVA; Accept: $F_{(75)} = 7.0$, $p<0.001$; Skip: $F_{(75)} = 2.6$, $p<0.001$; Each time bin via one-sided Wilcoxon Signed-rank vs 0 corrected for multiple comparisons; ANOVA grouped by subregion; Accept: $F_{(3)} = 45.6$, $p<0.001$; Skip: $F_{(3)} = 49.5$, $p<0.001$). This was the case for both accept and skip decisions and throughout the full decision period, indicating that rats maintained an understanding of their current position within the task regardless of what offer they received or how they responded to it. In stark contrast, neural coding for the delay offered (cost) was absent across all four mPFC subregions (*Figure 5D*; Rm ANOVA; Accept: $F_{(75)} = 2.1$, $p<0.001$; Skip: $F_{(75)} = 1.3$, $p<0.05$; Each time bin via one sided Wilcoxon Signed-rank vs 0 corrected for multiple comparisons, no time bins reached significance, all $p>0.05$; ANOVA by subregion; Accept: $F_{(3)} = 2.2$, $p=0.09$; Skip: $F_{(3)} = 1.2$, $p=0.31$). To verify that any conjunctive representation of reward rank was not obscuring our ability to detect delay coding in mPFC, we repeated this analysis independently for each restaurant rank (*Figure 5—figure supplement 2*). Even accounting for potential multiplexed coding, we did not detect significant representations of delay during the OZ. Fascinatingly, although mPFC (particularly ACC and dPL) encoded the decision made, the firing rates of mPFC cells never consistently represented the quality of the offer that was being considered. This finding suggests that effective decision making in the Offer Zone cannot involve mPFC operating in isolation, but rather must require active communication with other brain area(s) that do contain the necessary information about what delay has been presented.

One potential consideration in our reported results is the lower sampling of ACC cells relative to other subregions. However, we note that our findings of ACC involvement in decision making is consistent with the broader literature (*Hyman et al., 2012*; *Hyman et al., 2013*; *Ito et al., 2015*; *Lapish et al., 2008*; *Sul et al., 2010*) and that the bias that we observe for ACC engagement during skip decisions is in line with a proposed role of this structure in bypassing prospective rewards and searching out alternatives (*Brockett et al., 2020*; *Hayden et al., 2011*; *MacDonald et al., 2000*; *Mashhoori et al., 2018*; *Shenhav et al., 2013*; *Tervo et al., 2021*). To directly test the potential consequences of limited sampling on our reported results, we repeated all analyses after subsampling our recordings of dPL, vPL, and IL to match those of ACC (*Figure 5—figure supplement 3*). Importantly, the qualitative nature of all results held, though predictably the statistical power was reduced. Therefore, while sampling bias from ACC and insufficient statistical power remain possibilities, our findings of mPFC responding in the OZ are largely robust.

## Re-evaluative decisions are represented in the mPFC

While these rats made most of their decisions in the Offer Zone, after entering the Wait Zone rats are free to re-evaluate their decision and quit the offer countdown. Furthermore, multiple manipulation studies have found differential effects on decision making in the OZ vs in the WZ (*Sweis et al., 2018b*; *Sweis et al., 2018c*), presenting a unique opportunity in our task to examine a distinct secondary decision process. We directly measured the neural coding for the decision in the WZ (either quit or earn) by computing the mutual information between neural firing rates and the eventual choice. At the beginning of the WZ, there was no coding for choice across any of the four subregions (*Figure 6A*; Repeated measures ANOVA for subregions over time bins; Interaction: $F_{(207)} = 1.3$, $p<0.01$; Post-hoc

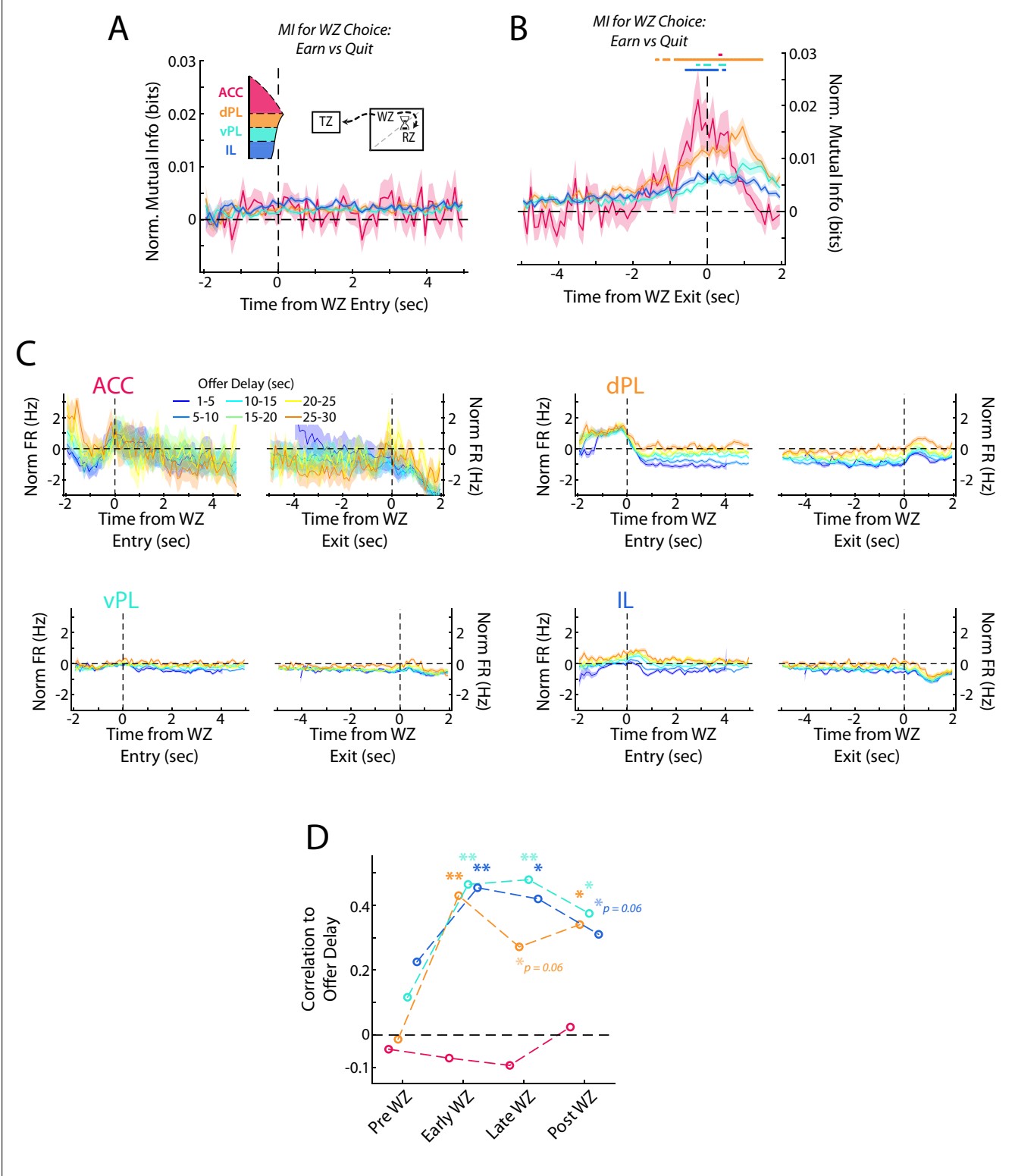

**Figure 6.** Re-evaluative decisions are represented in mPFC. (**A, B**) Mutual information (MI) between firing rate and WZ choice (earn vs quit) for cells in each mPFC subregion at the beginning and end of delay period. Data are show in (**A**) aligned to WZ entry and in (**B**) aligned to WZ exit. Time bins with a significant MI response (one-tailed Wilcoxon Signed-rank, α=0.05 after multiple comparison correction; n=70 time bins) are identified by colored bars above each plot. (**C**) Average firing rate of mPFC cells for each subregion separated by the offer delay that had been accepted. Data are aligned to the beginning of the WZ and the end of the WZ. (**D**) Correlation between average firing rate across mPFC cells and the offer delay that had been

*Figure 6 continued on next page*

*Figure 6 continued*

accepted. Average firing rate profiles were computed across all cells for each rat independently in each offer delay bin (up to 48 values: 8 rats x 6 delay bins). These firing rates were then correlated against offer delay during four time windows: Pre WZ, Early WZ, Late WZ, and Post WZ. See Methods for additional details. For (**A–C**), data are shown as the Mean ± SEM across cells in each mPFC subregion. All data were normalized against analyses performed using shuffled spike times.

The online version of this article includes the following figure supplement(s) for figure 6:

**Figure supplement 1.** Responses in the Wait Zone from subsampled data.

**Figure supplement 2.** Net firing rate between Quit and Earn responses.

**Figure supplement 3.** Similarity of quit decisions and leaving after reward.

on each time bin via one sided Wilcoxon Signed-rank vs 0 corrected for multiple comparisons, no time bins reached significance, all p>0.05). However, examining neural activity at the end of the WZ, just before either the quit decision was executed and the rat left the WZ or the delay ended and food was delivered, we found a significant increase in mutual information for the re-evaluative choice made (*Figure 6B*; Rm ANOVA; Interaction: F(207) = 4.4, p<0.001; Each time bin via one-sided Wilcoxon Signed-rank vs 0 corrected for multiple comparisons). While a significant increase occurred across all four subregions, the magnitude of this increase was stronger in dorsal mPFC (ACC and dPL) than in ventral mPFC (vPL and IL) (ANOVA grouped by subregion; F(3) = 17.3, p<0.001; Mann-Whitney U test, dmPFC (ACC & dPL) vs vmPFC (vPL & IL); z=4.2, p<0.001). Furthermore, the increase occurred earliest and persisted for longest in the dPL subregion, suggesting particularly strong coding for re-evaluative decision making in dPL. Interestingly, although ACC activity in the OZ coded the primary decision more strongly than was detected in other prefrontal areas, this was not the case for re-evaluation decisions in the WZ where the magnitude of MI change was comparable between ACC and dPL (ANOVA HSD post-hoc ACC vs dPL, p=0.93). Although we sampled fewer ACC cells, down-sampling of data from the other subregions yielded qualitatively similar results to those from our full data set (*Figure 6—figure supplement 1*). It is therefore likely that while we were unable to detect significant responses in ACC early in the decision process, the qualitative response pattern we observed is an accurate representation of the subregion's activity pattern. Collectively, these results indicate that while coding for re-evaluative decisions was apparent across the full mPFC, it exhibited a gradient behavior with more prominent responses observed in the dorsal areas.

In the RRow Offer Zone, choices are registered based on a motor output either to the Wait Zone (Accept) or the Transition Zone (Skip). In the Wait Zone, the decision entails either passively remaining at the location (Earn) or actively leaving the site (Quit). To determine if our neural response in the WZ may be more indicative of a motor response, we computed the net firing difference between earn and quit events for each cell (*Figure 6—figure supplement 2*). Importantly, we found cells with firing rate biases for both options, as would be expected if the decision process were driving the neural activity. Additionally, we found that ACC cells trended toward being more likely to increase their firing rate for a quit decision, or one in which rats bypassed available food for an alternative (64% net Quit; $X^2$=2.9, p=0.08). This mirrors both our findings in the OZ (*Figure 5B*), and past work on ACC (*Hayden et al., 2011*; *Mashhoori et al., 2018*; *Tervo et al., 2021*).

As in the OZ, decision making in the WZ relies on the evaluation of the delay to be waited and a determination of whether or not that time expenditure is advantageous. In the OZ this information was surprisingly absent from representations in mPFC (*Figure 5D*), pointing to the need for inter-regional communication. However, after rats entered the WZ, firing rates in dPL, vPL, and IL all varied systematically according to the offer delay that had been accepted, with higher firing rates associated with longer delays (*Figure 6C and D*). Notably, no relationship existed for the ACC subregion, the area with the strongest coding for the initial choice (ACC: p>0.62 at all time points). Although it remains possible that our limited sampling of ACC prevented detection of a relationship between firing rate and delay, or that sampling biases might have skewed our findings, down-sampling our recordings from dPL, vPL, and IL to a comparable size did not eliminate the effects seen elsewhere (*Figure 6— figure supplement 1*). Consistent with our mutual information results, no significant correlation existed between average firing rate and delay in the 1 s period prior to WZ entry while rats were still in the OZ (Pre WZ: all p>0.19). However, at both the beginning and the end of the WZ there was a significant correlation between firing rate and delay, indicative of coding for the value of the accepted

offer (Early WZ: dPL, vPL, IL all p<0.01; Late WZ: dPL p=0.06, vPL and IL p<0.01). Notably, this coding for offer value persisted past the end of the delay period (Post WZ: dPL and vPL p<0.05, IL p=0.06) indicating that although value coding was not present within mPFC while the initial OZ decision was being made, it strongly developed across multiple subregions during the re-evaluation period.

## Responses to reward delivery differ across mPFC subregions

After accepting the offer and waiting out the full delay, rats in RRow earn a food reward. This event provides both a highly salient natural reward, and also serves as a learning signal for the rats to determine if their past decisions were a worthwhile use of the time expended. We examined mPFC activity patterns at the beginning of the Reward Zone as reward was delivered and observed qualitatively different neural responding across the four subregions (*Figure 7A and B*, *Figure 7—figure supplement 1*; ANOVA grouped by subregion; F(3) = 13.1, p<0.001). ACC activity was characterized by a strong decrease in firing rate that persisted for several seconds. Across the remaining three subregions the responses were heterogenous across the populations in each area, but the average responding varied from dorsal to ventral with dPL exhibiting a net increase, IL a net decrease, and vPL falling into a middle ground with balanced increases and decreases in firing rate. Interestingly, responding to reward delivery tended to occur sooner in the ventral portions of mPFC, though across the population there was considerable variation (*Figure 7C*; ANOVA grouped by subregion; F(3) = 10.5, p<0.001; Mann-Whitney U test, dmPFC (ACC & dPL) vs vmPFC (vPL & IL); z=4.8, p<0.001).

After earning a reward, rats then must advance to the next restaurant in order to receive a new offer and earn more food. However, this advancement is un-cued and entirely at the volition of the rat. Interestingly, rats tended to spend considerable portions of their total time lingering in the RZ, consuming their earned food and waiting to proceed to the next site (*Figure 1—figure supplement 1*), consistent with prior observations of rats and mice on this task (*Schmidt and Redish, 2021*; *Steiner and Redish, 2014*; *Sweis et al., 2018a*; *Sweis et al., 2018c*; *Sweis et al., 2018d*). Because this lingering time is self-directed and presents a limiting factor to earning more food, we hypothesized that neural activity during this time may provide additional insight into subjective decision making. Indeed, we found that firing rates within mPFC subregions systematically varied according to the lingering behavior of the rat (*Figure 7D and E*).

Interestingly, when examining the first half of the lingering period, we found significant relationships in the ventral portions of mPFC, vPL and IL, between firing rate and eventual linger time (Pre RZ: all p>0.66; Early RZ: vPL p<0.05, IL p=0.07). However, in the second half of the lingering period, this relationship between firing rate and lingering time disappeared in the ventral mPFC and an opposing relationship developed in the dPL subregion (Late RZ: dPL p<0.01; Post RZ: all p>0.39). While we did not observe any systematic effect in the ACC subregion, this may be attributable to the limited sampling or potential biases within our data set. Notably, relationships between firing rate and linger time in the other three subregions were obscured when down-sampling the total population (*Figure 7—figure supplement 1*). Together, these findings suggest the potential for multiple cognitive processes during the lingering period and a gradual shift of neural engagement along the dorso-ventral gradient of the medial wall. During the early portion of the lingering time, when rats were presumably processing the fact that they had received food of a specific flavor, ventral portions of mPFC were more strongly engaged, potentially related to motivational or learning factors. During the later portion of the lingering time, when rats were considering moving on to the next foraging site, dorsal portions were more strongly engaged, potentially related to the more acute decision of specifically taking the action to transition to the next restaurant.

## Discussion

Clear functional distinctions exist along the medial wall of the prefrontal cortex. Our results identified four subregional units that closely match well-established anatomical designations, thus presenting a strong correspondence between the functional and anatomical understanding of the mPFC. Examination of the activity of each of these subregions during complex decision-making further revealed subtle but significant differences in their information processing and pointed to a gradient of function across the prefrontal cortex. While each prefrontal subregion was engaged across a wide array of functional components of decision making, a biasing emerged between the dorsal and ventral ends

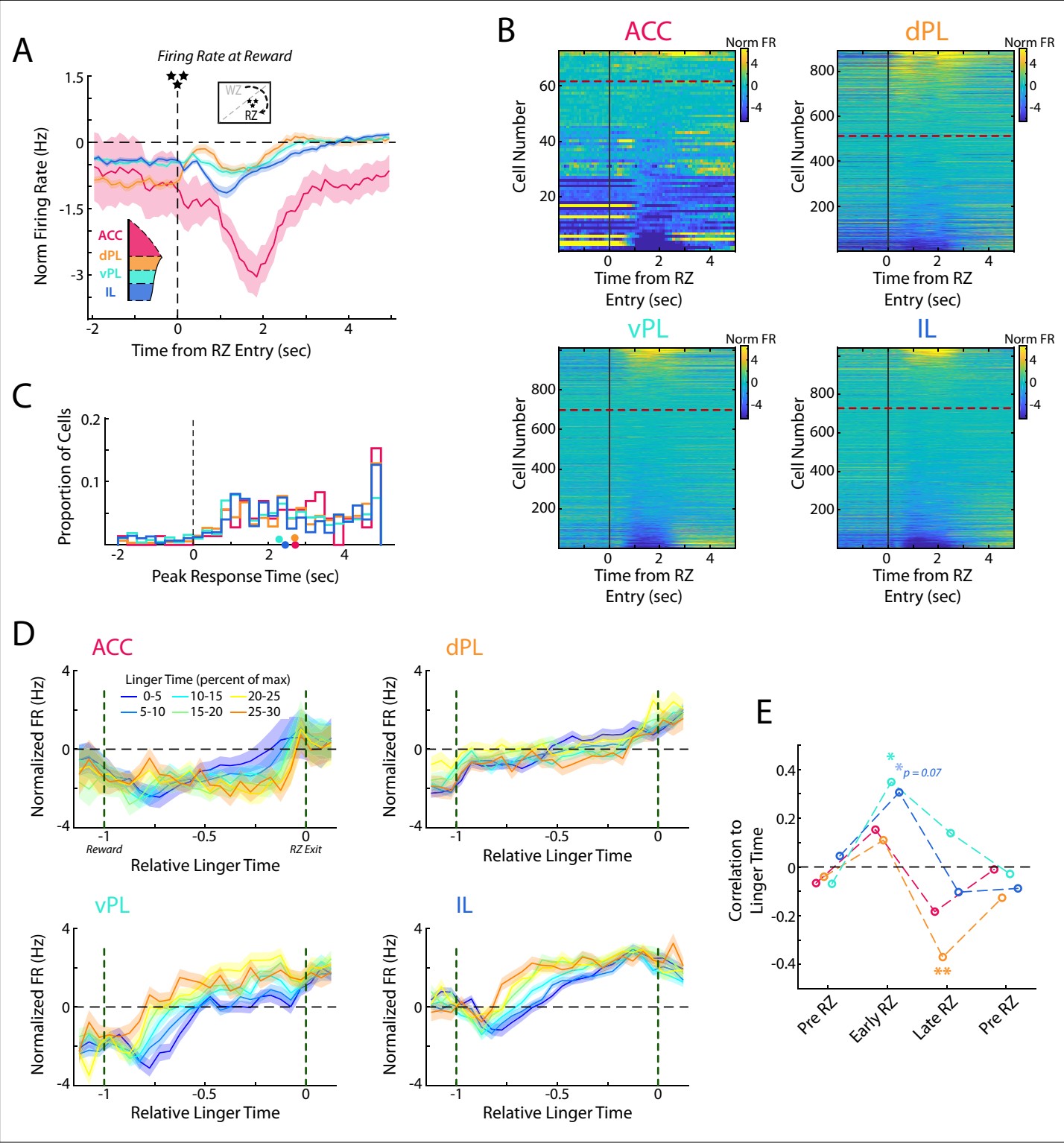

**Figure 7.** Responses to reward delivery differ across mPFC subregions. (**A**) Average firing rate for cells in each mPFC subregion as rats entered the Reward Zone and received their reward. (**B**) Distribution of all cell responses for each subregion at RZ entry. Cells are sorted according to the average activity from 0.5 to 3 s after RZ entry. (**C**) Time of the peak cellular response in the first portion of the RZ period (−2–5 s from RZ entry). Distributions of all cells are shown as histograms and mean response times for each subregion are indicated with a colored dot. (**D**) Average firing rate of mPFC cells for each subregion separated by the amount of time spent in the RZ (linger time). Data are grouped according to the percentage of maximal linger time computed independently for each session. Note that duration in the RZ is normalized across laps to produce a relative duration for analysis. (**E**) As for

*Figure 7 continued on next page*

*Figure 7 continued*

*Figure 6D*, correlations between average firing rate across mPFC cells and the linger time percentage. Average rate profiles were computed for each rat in each linger time bin (up to 48 values) and correlated to percentage of linger time. Average firings rates and correlations were computed in four time windows: Pre RZ, Early RZ, Late RZ, Post RZ. See methods for additional details. For (**A, D, E**), data are shown as the Mean ± SEM across cells in each mPFC subregion. All data were normalized against analyses performed using shuffled spike times.

The online version of this article includes the following figure supplement(s) for figure 7:

**Figure supplement 1.** Responses in the Reward Zone from subsampled data.

of prefrontal cortex. Neural activity in dorsal portions of the mPFC was more strongly engaged during acute decision making, both in the Offer Zone where initial value-based choices were made, and in the Wait Zone where choices were re-evaluated. In conjunction, ventral portions of mPFC were more strongly associated with variables that might be linked to more motivational factors such as time into the behavioral session, and how long to linger after receiving food before electing to pursue the next reward.

## Prefrontal functional divisions mirror anatomical divisions, with an added segregation of the prelimbic cortex

The medial prefrontal cortex has long been attributed a role in a diverse array of functions (*Dalley et al., 2004*; *Euston et al., 2012*; *Miller and Cohen, 2001*) and from an anatomical standpoint it has long been known that different portions of the medial wall have differing connectivity profiles (*Groenewegen and Uylings, 2000*; *Kolb, 1990*; *Uylings et al., 2003*). Although these anatomical differences imply that the ACC, PL, and IL regions likely also differ functionally, no single study had yet fully addressed this question, meaning that to draw any conclusions about functional differences between mPFC subregions requires integration across different experimental approaches (*Carlén, 2017*; *Kesner and Churchwell, 2011*; *Laubach et al., 2018*; *McLaughlin et al., 2021*; *Miller and Cohen, 2001*). Furthermore, any direct correspondence between functional and anatomical domains has thus far remained elusive. Using the precise anatomical localization afforded by silicon probes and the diverse engagement of mPFC provided by Restaurant Row, we discovered that functional communication within the mPFC occurs in a localized pattern consisting of four distinct processing subregions. Furthermore, the anatomical boundaries between these processing hubs closely match

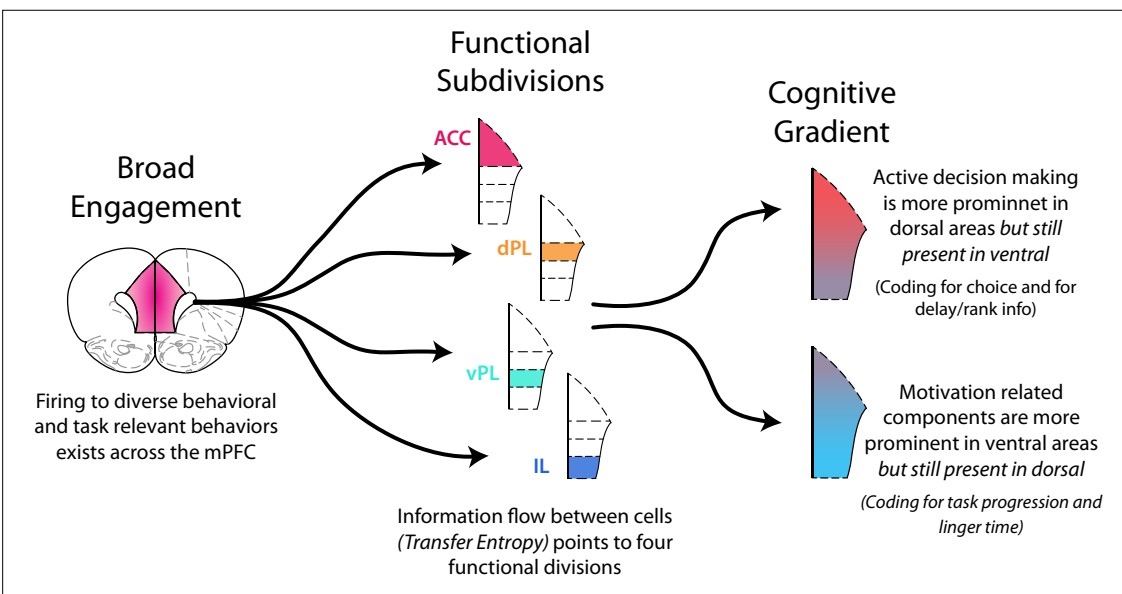

**Figure 8.** Functional activity in the medial prefrontal cortex. Functional activity in mPFC can be described in multiple ways. There was broad, distributed engagement across subregions for many behavioral variables. Analysis of functional coupling between cells using TE revealed distinct subregions. In many cases functional activity exhibited a gradient along the dorso-ventral axis. Neural engagement during active decision processing was more prominent in dorsal regions while motivation related components were more strongly represented in ventral areas.

anatomically defined subregions, highlighting a tight coupling between the functional and anatomical domains.

However, the emergent story is more complex than the standard anatomical understanding of mPFC as composed of anterior cingulate, prelimbic, and infralimbic cortices. In addition to the boundaries between these three subregions, we also report evidence for functional division of the PL into dorsal and ventral subregions. While there is anatomical data supporting a division of PL into dorsal and ventral parts (*Berendse et al., 1992*; *Heidbreder and Groenewegen, 2003*; *Vogt and Paxinos, 2014*; *Voorn et al., 2004*), our work shows that they process information differently as well.

## Distributed vs Discrete vs Gradient: Functional activity in the prefrontal cortex

What is the best way to characterize functional activity throughout the mPFC? Some reports of prefrontal activity have described distributed representations where many cells across the cortex are all engaged in a given functional process, or in turn where the activity of each cell multiplexes information across a large set of multiple processes (*Duncan, 2001*; *Rigotti et al., 2013*; *Seamans et al., 2008*; *Stokes et al., 2013*; *Wallis et al., 2001*). In some aspects of mPFC function we find a similar picture (*Figure 8*). For each behavioral variable we examined, we found it to be represented by a distributed population of cells spread across all four prefrontal subregion (*Figure 2*). Each individual cell also responded to multiple different variables as would be expected in a distributed coding scheme.

Alternatively, one can think of discrete units of functional processing within mPFC with each component primarily involved in a particular function. The idea of distinct component of mPFC is strongly rooted in anatomical work that points to the prefrontal areas of anterior cingulate, prelimbic (dorsal and ventral), and infralimbic cortices (*Groenewegen and Uylings, 2000*; *Heidbreder and Groenewegen, 2003*; *Hoover and Vertes, 2007*; *Kolb, 1990*; *Sesack et al., 1989*; *Uylings et al., 2003*). In parallel, many functional studies, particularly those that seek to identify functional dissociations between subregions, have worked to attribute distinct cognitive processes to each prefrontal subregion (*Burgos-Robles et al., 2013*; *Capuzzo and Floresco, 2020*; *Dalton et al., 2016*; *Killcross and Coutureau, 2003*; *Mukherjee and Caroni, 2018*; *Ragozzino et al., 1998*; *Sierra-Mercado et al., 2011*; *Vidal-Gonzalez et al., 2006*). Notably, our measurement of functional communication within the mPFC using TE points to the existence of four distinct processing units (*Figure 3*). Thus, in some respects mPFC appears to be composed of distinct functional subunits (*Figure 8*).

Finally, activity within the prefrontal cortex could be thought of as a gradient of functional processing. Indeed, many aspects of our data point to this perspective. Dorsal aspects of mPFC (ACC & dPL) were strongly engaged while rats were making their decisions (*Figures 5 and 6*); however ventral aspects (vPL & IL) also represented these same decision components, but to a lesser degree. In parallel, neural activity in ventral aspects was strongly associated with motivational factors (*Figures 4 and 7*), but dorsal cells also represented this information. While all aspects of mPFC were engaged in each of the functional processes we examined, the degree of engagement was non-uniform. Biases existed and a functional gradient emerged in moving from dorsal to ventral along the medial wall (*Figure 8*).

The combination of these three perspectives points to a nuanced picture of mPFC function: Discrete subregions exist, which preferentially processes different aspects of the cognitive needs of a task (here RRow). But, these subregions, like the cells within them, multiplex the necessary cognitive information along a dorso-ventral gradient. By combining anatomical information, measures of network communication (such as through TE), and analyses of the relationships between neural activity and behavior (such as through mutual information), it becomes possible to determine how complex brain regions, such as the mPFC, process information. Future studies should build on this work and examine the consequences of manipulating these functional relationships and the causal roles of the distinct mPFC subregions in learning and action selection. Complex tasks such as RRow provide a particularly useful avenue for such study as they provide multiple access points for teasing apart the complex functions of prefrontal cortex (*Schmidt and Redish, 2021*; *Sweis et al., 2018b*).

## Restaurant Row as a window into decision making

Decision making is an inherently complex and interconnected topic. One method for addressing this complexity entails isolating a particular component of the decision process in a specifically designed behavioral task. Here, we took an alternative approach that built on the complexity of the naturalistic behavior available in the Restaurant Row task (*Steiner and Redish, 2014*; *Sweis et al., 2018a*). Successful performance in RRow requires monitoring and integrating a wide swath of relevant variables, all of which we found to be represented within the mPFC. But the task also presents a series of independent and interacting decision processes. The Offer Zone and Wait Zone have been linked to distinct decision processes, with experimental manipulations affecting decisions in each independently (*Sweis et al., 2018b*; *Sweis et al., 2018c*). Our data reinforce this understanding with functional differences between ACC and dPL between these task phases. Manipulations specifically of the dorsal mPFC have been found to alter interactions with hippocampus and signatures of deliberative strategies in multiple planning stages of this task (*Schmidt et al., 2019*; *Schmidt and Redish, 2021*). Here again our data inform this past work, finding that choice coding preferentially emerged within these same dorsal areas of mPFC that are interconnected with areas such as hippocampus that are implicated in deliberation. Thus, Restaurant Row provides an important middle ground in the study of decision making. It provides a scenario that mirrors decision making in the natural world and engages a wide array of neural processes operating in concert, while at the same time providing instances of reliable and well-controlled windows into specific decision-making processes for study with causal manipulations or neural recordings.

The foraging economics of RRow parallels similar results seen in other rodent economic tasks. Another study that directly sought to answer how prefrontal firing is involved in economic decision making is the work of *Sul et al., 2010*. Using a bandit task, this study found that firing across the prefrontal cortex, both medial prefrontal (ACC and PL/IL) and orbitofrontal (OFC), represented the identity of an upcoming decision just prior to rats making their choice. Furthermore, this coding for choice was stronger in ACC as compared to PL/IL, a finding that matches our results. When examining neural signals of value, Sul et al. found limited coding within the ACC and PL/IL subregions with stronger representations in OFC. Here again our results match this past work as we find no evidence for coding of value (here being offer delay) while rats make their initial value decision in the OZ.

One limitation that exists in many naturalistic decision-making tasks is the ability of an experimenter to distinguish between neural activity associated with a decision process as compared to a motor output. Externally, it is only possible to gauge that a decision has been made based on an animal's motor outputs (or lack of outputs), intrinsically linking the two processes. While naturalistic decision tasks cannot escape this reality, RRow provides two cases that can begin to dissociate between decisions and motor actions. First, decisions in the Offer Zone entail prescribed directions of movement: accept decisions are always left turns and skip decisions are always right turns. Accordingly, examining neural activity either ipsilateral or contralateral to the direction of movement can provide clues of what may be related to motor output vs decision (*Figure 5—figure supplement 1*; *Erlich et al., 2011*). Second, decisions in the Wait Zone require rats either to actively leave the site (quits) or to passively remain (earn). Thus, if neural activity is associated with motor responding one should observe firing increases only during active quit decisions. In our data, we found firing rate increases during both responses, pointing toward engagement of mPFC in the decision component (*Figure 6—figure supplement 2*). Additionally, we can compare neural response when rats quit a decision in the Wait Zone and when they eventually leave the Reward Zone after consuming their food (*Figure 6—figure supplement 3*), as both cases entail similar motor actions, but reflect markedly different cognitive states. While there were correlations between the neural responses between these two periods, especially in the more dorsal aspects of mPFC, firing of the majority of cells differed markedly between quitting an offer and leaving the reward zone after consuming a reward. Thus, our data suggest that in RRow, decision processing is a stronger driver of mPFC activity than motor output.

## Prefrontal cortex and its involvement in decision making

Current theories suggest that a central purpose of the mPFC is to support effective decision making as organisms interact with their world (*Dalley et al., 2004*; *Euston et al., 2012*; *McLaughlin et al., 2021*; *Miller and Cohen, 2001*). Building off of this framework, our findings can provide valuable insight into how the different components of mPFC may operate in concert to support decision-making,

particularly in naturalistic foraging scenarios such as RRow (*Figure 8*). We found that each task-relevant variable that we examined was represented within the neural firing patterns of mPFC to a significant degree, presumably to be employed in furthering the rat's goals of understanding their world and securing rewards. Yet information was not represented uniformly across the full mPFC. We found that two distinct cognitive domains emerged, and that these were anatomically biased between the dorsal and ventral aspects of mPFC. In the dorsal mPFC (ACC and dPL), variables related to active decision making were most prominent, and strong changes to firing patterns occurred at critical decision points in the task. In contrast, neural activity in the ventral mPFC (vPL and IL) was more related to task variables that changed slowly across the session, such as motivation-related factors. Interestingly, this functional biasing between dorsal and ventral mPFC has a strong anatomical basis (*Heidbreder and Groenewegen, 2003*; *Voorn et al., 2004*): dorsal components of mPFC more strongly project to structures such as dorsal hippocampus, dorsal striatum, and premotor cortex which are collectively involved in the selection and execution of specific actions (*Heidbreder and Groenewegen, 2003*; *Heilbronner et al., 2016*; *Hoover and Vertes, 2007*; *Sesack et al., 1989*), while ventral mPFC has more connectivity with ventral hippocampus, ventral striatum, amygdala, and hypothalamus which are more tied to motivation components and generalized status monitoring (*Heidbreder and Groenewegen, 2003*; *Heilbronner et al., 2016*; *Hurley et al., 1991*; *Sesack et al., 1989*).

Moving beyond this initial perspective of a functional gradient from dorsal to ventral mPFC, our data highlight differences across the four subregions. ACC was more strongly engaged during Offer Zone decisions with notable weaker choice coding and TE signals in the Wait Zone. Comparatively, dPL was strongly engaged in the Wait Zone period where firing rate differences reliably differentiated earn vs quit decisions and encoded information about offer delay required in making these decisions. Therefore, our data suggests a bias toward ACC engagement during primary decision making (Accept vs Skip in the Offer Zone) with dPL then engaged during re-evaluation of these decisions (Earn vs Quit in the Wait Zone). In the ventral mPFC, we found that neural activity in both vPL and IL were strongly correlated to motivation-related components of the task (task progression and the first part of the linger time). While analysis of real-time coupling between cells clearly separated the two ventral subregions, further analysis in the task found only limited differences between them. One possibility is that this lack of dissociation between vPL and IL is a product of RRow as a task where overall motivational factors are less critical for success than the active decision making that drives dorsal mPFC. Future studies should specifically aim to parse out any functional differences in ventral mPFC through directed examination of motivational components on decision making.

## Methods

**Key resources table**

| Reagent type (species) or resource | Designation | Source or reference | Identifiers | Additional information |
|---|---|---|---|---|
| Strain, strain background (*Rattus norvegicus*, 4 M, 4 F) | Fisher-Brown Norway F1 Hybrid | Bred in House | | Aged 9–15 months |
| Chemical compound, drug | Isoflurane | UMN Research Animal Resources (RAR) | | |
| Chemical compound, drug | Carprofin | UMN RAR | | |
| Chemical compound, drug | Dual-Cillin | UMN RAR | | |
| Chemical compound, drug | Baytril | UMN RAR | | |
| Chemical compound, drug | C&B Metabond | Patterson Dental | Cat #: 553–3484 | |
| Chemical compound, drug | Pentobarbitol | UMN RAR | | |
| Chemical compound, drug | Paraformaldehyde | Sigma-Aldrich | Cat #: 158127 | |
| Software, algorithm | Matlab v2017b | Mathworks | RRID: SCR_001622 | For Formal Analysis |
| Software, algorithm | Matlab v2015b | Mathworks | RRID: SCR_001622 | For Running Task |

*Continued on next page*

*Continued*

| Reagent type (species) or resource | Designation | Source or reference | Identifiers | Additional information |
|---|---|---|---|---|
| Software, algorithm | Kilosort v2.0 | Marius Pachitariu | https://github.com/MouseLand/Kilosort (*Pachitariu et al., 2016*; *Stringer et al., 2019*) | Downloaded Jan 2020 |
| Software, algorithm | Phy v2.0 Beta 1 | Cyrille Rossant | https://github.com/cortex-lab/phy (*Rossant, 2022*). | Downloaded Feb 2020 |
| Software, algorithm | Analysis Code | This Paper | https://osf.io/s5xqm/ | Code written for the study |
| Other | Physiology and Behavioral Data | This Paper | https://osf.io/s5xqm/ | Data collected for the study |
| Other | Food Pellets (45 mg, 5TUL) | TestDiet | Cat #: 1811155; 1811443; 1812298; 1811645 | Reward food pellets; Flavors: Plain; Chocolate; Fruit "Cherry"; Banana |
| Other | 64 Ch Silicon Probe | Cambridge Neurotech | ASSY-156-H3 | Recording probe; H3 Model |
| Other | 256 Ch Intan RHD Recording System | Intan | Cat #: C3100 | Recording Hardware |
| Other | 64 Ch Intan RHD Headstage | Intan | Cat #: C3325 | Recording Headstage |

## Subjects

Eight adult Fisher-Brown Norway F1 hybrid rats (4 M, 4 F) were used in the study. Sample size (n=8 rats) was based on past studies and did not employ a formal power test. All rats were bred in house and at age 9–15 months during experiments. Rats were single housed in a temperature-controlled colony room with a 12 hr light/dark cycle (lights on at 8:00 AM). Throughout the experiment, rats were food restricted to at or above their 80% free feeding weight, except for approximately one week prior to implantation of silicon probes during which rats were given full access to food. All training and testing sessions were conducted during the light phase and each rat performed the behavioral task at approximately the same time each day. All procedures were approved by the University of Minnesota Institutional Animal Care and Use Committee (IACUC) and were performed following NIH guidelines.

Each subject was treated independently to yield a sample of eight biologic subjects contained within the study. No formal replication was performed, though the behavioral data reported here largely replicate previously published works.

## Restaurant Row task

Prior to all experiments rats were handled daily to acclimate them to human contact. They were then placed on food deprivation and for four successive days were offered 1 hr of access to flavored food pellets (45 mg/pellet, TestDiet, Richmond IN). After familiarization with the available food rewards rats were trained on the economic decision task Restaurant Row (*Steiner and Redish, 2014*; *Sweis et al., 2018a*). All behavioral events were operated through custom built Matlab software (Matlab v2015b, Mathworks) to track rat position, control task state, play tones, and trigger reward delivery. Video was collected via an overhead camera sampling at 30 Hz and tracking an LED that was attached to a backpack strapped to the rats (pre-op) or mounted to the implant (post-op). Tones were played on a set of external speakers and food delivery was via a Matlab-Arduino interface (Arduino Uno) sending TTL pulses to Med-Associate feeders (Fairfax, VT).

In Restaurant Row, rats had 1 hr each day to run clockwise around a raised octagonal track (130cm x 130cm overall footprint) and collect flavored food pellets at four reward sites (restaurants) located at the end of radial spokes. At each reward site, automated feeders delivered two reward pellets (90 mg total) of a unique flavor: cherry, banana, plain (unflavored), or chocolate. All food pellets were nutritionally equivalent and varied only in their flavor. Throughout the entire experiment flavors were fixed to their respective reward sites.

For each restaurant visit, rats first entered the Offer Zone (OZ) where the frequency of an auditory cue signaled the temporal delay that the rat was required to wait to earn a reward. Delays ranged from 1 s to 30 s, drawn pseudorandomly from a uniform distribution, with higher frequency tones corresponding to longer delays. A 1 kHz tone corresponded to the 1 s delay with 175 Hz steps up

in frequency for each additional second. Tones were played for 100ms and were repeated at 1 s intervals. While rats were in the OZ, tones continued to play at the same frequency, and time did not count down, until rats elected to either Accept the offer and move into the Wait Zone (WZ) or Skip the opportunity and advance to the next restaurant. Upon the rat accepting an offer and moving down the reward spoke into the WZ, the delay began to count down with tones of progressively lower frequency played each second. If the rat remained in the WZ until the delay reached zero, they Earned a reward and flavored reward pellets were delivered. Upon earning the reward rats were designated as now being in the Reward Zone (RZ) and could eat their reward and linger in the zone until ready to advance to the next restaurant. Note that the WZ and RZ were spatially coincident and were defined according to pre vs post reward delivery. However, at any time the rat could instead elect to Quit the offer, leave the WZ, and transition to the next restaurant. Upon exiting the WZ the offer was rescinded, tones stopped, and the reward could no longer be earned, but otherwise quit actions carried no additional punishment. After leaving a given restaurant (Skip, Quit, or Earn), rats proceeded through the Transition Zone (TZ) to the next site where a new delay offer would be presented, and a different flavor reward could be earned. Rats were permitted 1 hr of time on Restaurant Row each day, which represented their sole opportunity to receive food each day, presenting an inherent economic pressure.

## Restaurant Row initial training

Training on the Restaurant Row task occurred over the course of 25 days. Progressively longer potential delays were introduced over the course of training, but the task was otherwise identical to the full version. For training days 1–5 all offers were 1 s delays. For days 6–10 offer delays ranged from 1 to 5 s. Additionally, on days 1–7 a combined OZ/WZ was used in which all offers were automatically accepted, and rats simply had to remain within either zone to earn the reward. For days 11–15 offer delays ranged from 1 to 15 s. For days 16–25 delays spanned the full range of 1–30 s and was the full version of the Restaurant Row task. Two rats (R536/R543) received an additional 5 days of training in the final 1–30 s range. At the completion of this training sequence, rats were given free access to food for at least 1 week prior to surgery for implantation of silicon probes.

## Surgery

At the time of surgery rats were anesthetized with isoflurane (2% in $O_2$) and were placed into a stereotaxic frame. Carprofen (5 mg/kg, SC), dual-cillin (0.2 ml, SC), and saline (3 ml, SC) were administered for analgesia, prophylaxis, and hydration respectively (All drugs provided by UMN Research Animal Resources). A pair of ground screws were implanted over cerebellum and a 3D printed ring was attached to the skull via an anchor screw and C&B Metabond adhesive. Craniotomies were drilled over the mPFC (±0.6–0.7 ML, +2.5–3.0 AP) and either one or two (bilateral) linear 64-channel silicon probes (Cambridge Neurotech H3 model) were implanted targeting either the dorsal (ACC/PL) or ventral (PL/IL) extent of the PFC medial wall. Each probe had recording contacts spaced at 20 µm intervals spanning a total of 1260 µm along the dorso-ventral axis. For dual implants, one probe targeted dorsal while the other targeted ventral mPFC to span the full extent (~3 mm) of the medial wall. Targeting of dorsal and ventral mPFC on the left and right hemisphere was counterbalanced across rats, accounting for sex, but not considering any behavioral metrics.

Probes were housed in custom built 3D printed microdrives that allowed for independent adjustment of each probe along the dorso-ventral axis. After lowering the probes into brain (~1.5 mm; All recording sites into brain), craniotomies were filled with a sterile mixture of bone wax and mineral oil (3/1 by weight) and microdrives were adhered to the skull with Metabond. A 3D printed shell and head cap were attached to the skull mounted ring to protect the implant. Finally, an LED was attached to the rear of the shell to facilitate video tracking during recordings. At the completion of surgery rats were given an injection of Baytril (10 mg/kg, SC) and additional saline (20 ml/kg, SC) as well as oral administration of Children's Tylenol (0.8 ml).

## Restaurant Row post-op training

Rats were given three days of recovery following surgery and were then reintroduced to the Restaurant Row task for retraining. Retraining followed a similar sequence to original training but in an expedited fashion. Retraining lasted 10 days total: 1 day of 1 s, 2 days of 1–5 s, 2 days of 1–15 s, 5 days of 1–30 s. Additionally, for the final three days of retraining (1–30 s delays), rats ran the task

with the recording tether attached, familiarizing them with the process of performing the task under full task and recording conditions. After this 10 day retraining sequence rats continued to perform Restaurant Row during mPFC recordings for an additional 14 days (7 days for R506). Over the course of this retraining sequence, probes were gradually lowered to the appropriate depths targeting the dorsal (ACC/PL) or ventral (PL/IL) mPFC.

## Recordings

Recordings were performed over 14 days (7 days for R506) immediately following post-op retraining and positioning of the probes. A technical glitch during one session for R535 (day 5) precluded synchronization of neural data to behavior and was thus excluded. A total of 104 sessions of data were included across all analysis. The Restaurant Row task during recording sessions was identical to the full task used in training; rats were given 1 hr on the track and offer delays ranged from 1 s to 30 s sampled pseudorandomly from a uniform distribution. Before and after the 1 hr task period rats sat on a flower pot for 5 min to record baseline neural activity during quiet resting. These pre/post data were used in the process of single unit isolation but were otherwise not analyzed here.

All recordings were collected at a 30 kHz sampling rate using a 256 channel Intan RHD recording system connected to 64-channel Intan RHD headstages. An Arduino interface between Matlab and the Intan system facilitated time syncing of neural data to ongoing behavior and task events. Periodically throughout the 14-day recording sequence probes were lowered (35–150 µm) to increase the likelihood of recording a diverse array of units and to increase the overall sampling across the dorsoventral axis of the mPFC. All movement of probes occurred after completion of a day's recordings leaving 23 hr for probes to restabilize before the next recording session. While we took efforts to record from different cells across the experiment, we are unable to claim that all recorded cells are unique.

Single-unit identification was achieved using Kilosort v2.0 (https://github.com/MouseLand/Kilosort *Pachitariu et al., 2016*; *Stringer et al., 2019*) with post clustering curation performed manually using Phy (https://github.com/cortex-lab/phy). Each silicon probe was analyzed for units independently and included data from the 60 min behavioral session as well as the two 5 min pre/post periods. Prior to Kilosort processing, data underwent a median reference subtraction across the 64-channels of the given probe, and subsequent unit identification used a 600 Hz high-pass filter. Collectively we identified 4030 single units across all recording sessions. All isolation of single units was performed blind to the session's corresponding behavioral data.

## Histology

At the completion of all recordings, 3 s of 4µA of current was passed through 20 channels on each probe to induce gliosis at the final recording location. Two days later rats were overdosed with pentobarbital and were perfused transcardially with phosphate-buffered saline and paraformaldehyde. Probes were left in place for 3–5 hr and were then carefully removed from the brain, leaving a visible track of their final position. Brains were then extracted and post-fixed in paraformaldehyde for 24 hr before being transferred to a mixture of paraformaldehyde and sucrose for cryoprotection. Coronal sections (40 µm) through the mPFC were cut on a cryostat, mounted to slides, and stained with cresyl violet. For each probe, the entry point into the brain, final tip location, and trajectory were determined from serial sections. Together with records of experimenter movement of each probe through the brain, this information was used in localizing units in 3D space within mPFC (see below). Across all histological sections we compared the expected depth of probe tips based on movement records to the depth measured in processed tissue. We found that minor tissue shrinkage occurred during histological processing with depth in sections to be on average 92 ± 4% smaller than the in vivo placements. For two out of the 13 total probes, histology revealed the probe trajectory to be outside of mPFC (one in VO, and one in MO). Data from these two probes (1013 units) were excluded from all analyses leaving a total of 3017 units verified to be located in the mPFC. While a large proportion of ACC and dPL cells came from one rat each, for all analyses we verified that results were qualitatively similar across the remaining animals (data not shown).

## Data analysis

### Behavior

Each visit to a Restaurant Row restaurant was identified as a Skip, a Quit, or an Earn dependent on whether or not rats entered the WZ (Skip) and subsequently if they remined in the WZ throughout the full delay (Quit/Earn). For each visit, decision reaction time was taken as the elapsed time between entering and exiting the OZ either into the WZ in the case of accepting the offer or moving to the next restaurant for Skips. Quit time was the time spent in the WZ prior to a quit decision being made and rats exiting the WZ. Note that, by definition, for a given restaurant visit the quit time must be less than the accepted delay. Linger time was defined as the time spent in the RZ after delivery of the reward up to the point at which rats exited the zone. Offer decision thresholds were calculated for each restaurant (flavor) independently by the least-squares fit of a Heaviside function of receiving food at each visit as a function of offer delay. Note that both Quit and Skip instances were treated identically (no food earned). To calculate the consistency of thresholds across days, we correlated the four-element vector of restaurant thresholds between each pair of recording sessions, either both taken from the same rat (day-to-day consistency) or from different rats (null distribution). Threshold consistency was also computed as the variance for a given restaurant threshold across sessions as compared to the variance across restaurant thresholds for a given session. Reward rank was determined in each session by sorting restaurants by decision threshold with the largest threshold (the reward the rat was willing to wait longest for) ranked 1st. Ties between restaurant thresholds were broken by the total number of earned rewards with the greater number of earns identified as the more preferred restaurant. For some behavioral analyses, we computed the 'subjective value' of each offer delay as its signed distance from threshold. Note that thresholds were a function of both rat and restaurant. Good offers were those with a positive value and bad offers had a negative value. Progressive distance from zero reflected the magnitude of this offer quality.

$$SubjectiveValue = Threshold_{RR} - Delay$$

### Unit localization in standardized 3D space

To precisely evaluate functional activity along prefrontal medial wall every unit recorded from the mPFC was localized to its three-dimensional location in a standardized atlas space. This was achieved by integrating waveform profiles across probe recording channels, physical locations of these channels on the probe shank, daily records of turning depths of each probe through the brain, histological information about probe trajectories through the brain, and stereotaxic coordinates in a standardized atlas framework. First, for each cell we evaluate the 64 waveform profiles across all recording channels of a given probe and identified the channel with the maximal amplitude as the putative location of the cell body (*Csicsvari et al., 2003*). We then computed the linear distance of this channel from the ventral tip of the probe based on 20 μm spacing between recording channels. Using daily records of turning depth of the probe tip into the brain, we were able to identify of the depth along the probe trajectory of the identified recording channel and putative cell body location. Next, for the probe of interest we identified in histological sections the 3D location of the entry point of the probe into the brain and the final location of the probe tip. Note that because we only ever turned probes ventral, this histological position marks the maximal distance that the probe was turned into the brain. The relative location of these entry and end points were then identified in a standard Paxinos and Watson atlas (*Paxinos and Watson, 2007*). While our histological sections provided an AP resolution of 40 μm, identification of corresponding points in the atlas provided a much coarser assessment along this axis. Finally, we computed the location of our recorded cell in standardized 3D space by converting the relative distance of our putative cell body location along the probe trajectory in real space (histological and probe turn distance), to the relative position along the trajectory within atlas normalized space.

### Basic spiking and waveform properties

Average firing rate for each cell was defined as the total number of spikes detected over the full behavioral session divided by the 1-hr duration. This metric did not include spikes during the 5 min pre/post-recording periods. Waveforms for each cell were taken from the recording channel with the largest amplitude and thus inferred to be most proximally located to the cell body. The time

between the peak and valley of a cell's average waveform was taken as the spike width. Coefficient of variation for each cell was computed by binning spiking over the full 1-hr session into 50ms time bins, calculating the average firing rate in each bin, and dividing the standard deviation across these firing rates by their mean. For each cell, we calculated both the median inter-spike-interval (ISI) and standard deviation across the ISI during over the course of the 1-hr behavioral session. As a measure of bursting we computed for each cell the ratio between short ISIs (<10ms) and long ISIs (>125ms;~1 theta cycle). To distinguish between principal cells and interneurons, we compared average firing rate and spike width for our recorded cells. Data clustered into two largely separable groups and boundaries between them were drawn by hand by a trained experimenter.

## Normalization of Offer and Reward Zone time

Progression through both the Offer Zone and the Reward Zone were self-paced and varied across each restaurant visit. Therefore, we evaluated neural activity during these two periods with respect to the relative time spent in the zone. For each zone visit, we divided the total time spent in the zone into 20 bins such that each bin composed 5% of the total time. For the period before and after we took the time spent in the preceding zone and subsequent zone and divided it into 10 bins (10% relative time). Analyses included 3 bins pre and post the decision period for a total of 26 normalized time bins. This normalization method was used in *Figures 5 and 7D*.

## Peri-event time histograms (PETHs)

Peri-event time histograms (PETHs) were computed using standard methods over either normalized time bins (OZ and RZ) or absolute time bins (WZ). Analysis of the WZ used bin widths of 100ms. For each visit to a restaurant, we totaled the spike count in each time bin and divided by the bin width to compute average firing rate in each bin. We then computed the mean firing rate across all restaurant visits for each time point. To specifically examine firing profile differences across various task conditions additional PETHs were constructed in which only a subset of restaurant visits was included (e.g. only visits that led to a Skip decision, or only visits with an offer delay between 1 and 5 sec).

To account for chance behavior of mPFC spiking, not directly associated with ongoing behavior or task variables, we computed a series of PETHs based on permuted spike times of each cell. For each cell, the full spike train was circular shifted in time ('shuffled') by a random amount ranging from 1 min to 59 min to dissociate neural spiking from ongoing task events. PETHs were then calculated using the behavioral event times and the time shifted spike train ('shuffled PETH'). This process was repeated 30 times and the mean PETH across the distribution of shuffles was taken as the by chance, expected activity. For all analyses, PETHs of individual cells were normalized by subtracting this shuffled average.

## Mutual information

To directly relate the spiking of mPFC cells to behavior during decision making, we computed the mutual information (MI) between firing rate and specific task conditions (*Timme and Lapish, 2018*). Intuitively, MI quantifies how much information neural firing can provide in predicting a concurrent behavioral condition. For each behavioral variable data were binned into 5 bins of uniform width. For MI calculations over time, we followed a parallel approach to that of our PETH calculations in computing MI at each time bin treated independently. After calculating firing rates in each time bin (PETHs), firing rates were discretized into 5 uniformly spaced bins. For each time point, the binned spiking rates across all restaurant visits were collected along with the corresponding behavioral/ task condition data. The mutual information between the two matched vectors was then computed according to:

$$MI = \sum p\left(B, S\right) log_2 \left(\frac{p\left(B,S\right)}{p\left(B\right), p\left(S\right)}\right)$$

where **B** and **S** reflect binned behavior and spiking rate.

As for PETHs, we normalized our MI calculations by repeating all analyses after time shifting spike trains of our mPFC cells. Shuffling and MI computation was repeated 30 times for each cell and the mean across this shuffled distribution was taken as the chance level. All analyses subtracted this shuffled average from the MI computed on raw, unshuffled spike times.

## Correlation between firing and behavior

To relate firing rate of prefrontal cells to behavioral variables we computed the average firing rate in each task zone for each restaurant visit. For each behavioral variable, the corresponding value at each time point was taken and the Pearson's correlation was computed between the vectors of firing rate and behavioral values. To relate behavioral variables to each other, we used this same process but correlated a pair of vectors across different behaviors.

## Stepwise regressions between firing and behavior

To account for linear relationships between behavioral variables in the RRow task, we used a stepwise linear regression procedure to relate firing of mPFC cells to independent behavioral variables. We used two complementary methods for performing the stepwise regression but in both cases, we performed a sequential, stepwise linear regression to progressively remove the influence of a given behavioral variable on the neural firing before relating the next behavioral variables. That is, subsequent behaviors were fit to the residual firing rates after accounting for preceding behaviors.

In the first method, we fit each behavioral variable to mPFC spiking in a predetermined order of behaviors. Thus, each cell underwent an identical analysis and behavioral variables were regressed out in the same order across the full population. Order of behaviors was determined from the overall prevalence within the data set based on a standard correlation analysis (*Figure 2C*). In the second method, we used the Matlab function 'stepwisefit' which fits the stepwise regression for each cell independently. Regression order is determined by the strength of the regression for each individual, meaning that the order that each behavioral variable is regressed out will vary across the population. Note that in both cases, the 'Regression Rank' identifies the overall strength of the relationship to neural firing, not necessarily the original order in which it was applied in the analysis.

## Transfer entropy

To measure the degree to which the activity of mPFC cells may be informative about and/or influence each other, we computed the TE between the firing rates of pairs of simultaneously recorded mPFC cells (*Timme and Lapish, 2018*). Related to granger causality, TE measures the information provided by the past activity of an input (cell X) about the future of a response (cell Y) above and beyond the information derived from the response's own past. Note that TE is directional and that TE from X to Y does not necessarily equal the TE from Y to X. First, activity of all cells over the full Restaurant Row session was binned using 10ms time steps. For each pair of simultaneously recorded cells, we computed the TE between an input cell X and a response cell Y according to:

$$TE_{X \rightarrow Y} = \sum p\left(Y_t, Y_{t-1}, X_{t-1}\right) log_2 \frac{p\left(Y_t | Y_{t-1}, X_{t-1}\right)}{p\left(Y_t | Y_{t-1}\right)}$$

To normalize the TE values, we repeated the above analysis after time shifting spiking data for only the predictor X cell. The full spike train of the predictor X cell was circular shifted in time and then TE between the shifted predictor cell X and the unshifted response cell Y was calculated to yield a single by-chance TE value between the pair. This shuffling procedure was repeated 30 times to yield a distribution of chance TE values (30 chance values). A mean was then taken across these 30 shuffles and taken as the by-chance expected relationship between the pair of cells. This mean was subtracted from the raw calculation taken from unshifted spike times, and this net TE is reported.

To examine the relationship between pairwise TE and anatomical locations, we binned resultant TE values into a 2D histogram (spatial resolution of 100 μm) according to the DV position of the two cells included in the pair, paying mind to which was analyzed as the input cell X and which was the response cell Y. The mean TE was taken for each bin. TE was quantified at depths along the medial wall by averaging the normalized TE in bins along the identity diagonal. At each DV step, we calculated the mean across all bins within 200 μm; that is to say, bins in which both pairs of cells were within 200 μm of the dorso-ventral point of interest. Valleys in the quantification (subregion boundaries) were computed as the local minima of the quantification along the DV axis.

To calculate the TE values for activity in each task phase independently, we repeated the above procedures but only included time bins from when rats were located in the task phase of interest.

## Subsampling of Data in dPL, vPL, and IL

To account for the lower sampling of ACC cells relative to other subregions, we repeated main analyses after subsampling the populations of the dPL, vPL, and IL subregions to match that of ACC.

For TE calculations, we subsampled populations at random to match the cell count of ACC (72 cells) and repeated our quantification along the dorso-ventral depth and identification of subregional bounds as for the full data set. This subsampling procedure was repeated 500 times with subregional bounds computed for each iteration. Because TE is a pairwise metric, we also subsampled our data to match the average number of cell pairs formed by our ACC data set. For each other subregion we randomly sampled a number of cell pairs to match the number of within region pairs of our ACC data set (334 pairs) and the average number of across region cell pairs for our ACC data set (397 pairs). We then computed the subregional bounds on this subsampled data set and repeated the process 500 times. To compute distributions of TE across subregions, we used these same subsampling methods and report one iteration in the corresponding supplemental figure, though repeated sampling yielded comparable results.

For analyses of OZ, WZ, and RZ responding, we subsampled our populations of dPL, vPL, and IL cells to match that of ACC (72 cells). We then repeated our analyses for each subregion using only this subsampled data set. One such iteration is shown in figure supplements, although repeated iterations yielded comparable results.

## Changes in functional properties over the session

To test for any changes in functional properties over the session we divided our data into equal size blocks over the course of the session and repeated our TE analyses, and the main analyses in each of the OZ, WZ, and RZ. For TE calculations, the session was divided into 5 equal durations (12 min each, Blocks 1–5 from start to end of the session). Quantification of TE across depth, and average TE for within subregion cell pairs were calculated independently for each time block. For OZ, WZ, and RZ analyses the session was divided into 3 equal durations (20 min bins, Blocks 1–3 from start to end). For OZ and WZ the cumulative MI over the respective decisions periods was calculated in each block. For the RZ the average firing rate was calculated in the 2 s following reward delivery.

## Correlation between firing rates and offer delay/linger time

To relate firing of mPFC cells to either offer delay (WZ, *Figure 6*) or linger time (RZ, *Figure 7*), we computed the average firing rate profile for each subregion for each rat independently. For linger time, behavioral bin windows were taken as percentages from the minimum to the maximum linger time computed independently in each behavioral session. Note that because of the long tail in the distribution of linger times, 0–30% linger time (range in *Figure 7D*) accounts for 87% of all RZ visits. Average PETHs were computed for each behavioral bin (offer delay or linger time) yielding up to 48 total PETHs (8 rats x 6 bins) for each mPFC subregion. For offer delay behavioral bins correlations between these 48 PETHs and the respective behavioral bins were computed during four time windows. For the Wait Zone and offer delay analyses these were: Pre WZ (0–1 s before WZ entry), Early WZ (0–3 s after WZ entry), Late WZ (0–3 s before WZ exit), Post WZ (0–1 s after WZ exit). For the Reward Zone and linger time analyses these were: Pre RZ (3 bins before RZ enter), Early RZ (first half of RZ; 10 bins), Late RZ (second half of RZ; 10 bins), Post RZ (3 bins after RZ exit). For each time window, the average firing rate over the window was computed for each PETH and the Pearson's correlation was computed between firing rate and center of the behavioral bin.

## Statistics

Comparison of threshold variance across restaurants vs across sessions was performed with a paired Wilcoxon Signed-Rank test with the rats as the samples. Behavioral analysis of time in the offer zone (reaction time) between accept and skip decisions was through a paired Wilcoxon Signed-Rank test between average reaction time at each offer value (data points for each rat). This method accounts for any variation in reaction time that may be driven by the offer value itself and directly compares between the two decision processes. Correlations between behavioral times (reaction time, quit time, linger time) and offer value were similarly performed after averaging metrics at each offer value (data points for each rat).

All analyses of single-cell firing patterns were normalized by subtracting the mean of "shuffled" data generated by circularly time-shifting spiking with respect to behavior. Thirty iterations of each shuffle were performed and the mean was taken across the distribution. An identical procedure was used for pairwise TE analysis but only spiking from the input X cell was shuffled.

Comparison of cell spiking to behavior was made on the distribution of correlations for each behavior of interest. In each case, the raw correlations were compared against a distribution generated from the same behavioral data but using shuffled spike times using a Wilcoxon Signed-Rank test. Spike shuffling was repeated 30 times and the mean correlation across the shuffles was taken for each cell.

Comparison of overall TE metrics against chance were performed with a Wilcoxon Signed-Rank test of shuffle normalized TE values against 0. Note this is equivalent to a matched test between raw data and shuffled data. TE quantified along the DV axis was evaluated as multimodal via Hartigan's Dip Test (*Hartigan and Hartigan, 1985*). Comparison between TE within a subregion vs across subregions was performed with a Mann-Whitney U test. To evaluate TE as a function of distance between subregions, pairwise TE was grouped according to the number of steps between a pair of mPFC subregions (0 Steps for within subregion, 1 Step for adjacent regions, etc.). Comparison between step sizes was via an ANOVA and individual pairs of step sizes evaluated with Tukey-Kramer HSD post-hoc criterion.

For generalized task variables (Task Phase, Session Time, Reward Rank), mutual information values were computed for each cell and normalized as described above against shuffled spike times. Normalized mutual information values were identified as significant using a Wilcoxon Signed Rank test. Comparisons across mPFC subregions was via an ANOVA and HSD post-hoc testing.

Comparisons of mutual information metrics (OZ and WZ choice) over time were performed using a repeated measures ANOVA with time bins as the repeated variables. OZ comparisons used 26 time bins and WZ used 70 bins. Post-hoc comparisons for each subregion were computed using a series of Wilcoxon Signed-Rank tests comparing mutual information at each time point against 0. Tests were one-tailed in the positive direction and significance levels for these tests were evaluated after Bonferroni correction for multiple comparisons; n=26 or 70 time bins. To compare responses across subregions, we computed the average response for each metric over a specific period of interest to yield a single value for each cell. In the OZ this period was from the OZ entry through the end of the analysis window (end of post-OZ). For the beginning and end of the WZ, this period was the first 2 s after WZ entry or the last 2 s before WZ exit. A comparable method was used for summarizing average responses of mPFC firing rates in response to reward delivery. In the RZ, the period of interest was from 0.5 s after RZ entry to 3 s after RZ entry. Comparisons across mPFC subregions was via an ANOVA and HSD post-hoc testing. In cases of comparing dorsal mPFC (ACC and dPL) vs ventral mPFC (vPL and IL) we used a Mann-Whitney U test.

To compare proportions of mPFC cells that were biased toward accept vs skip decisions in the OZ, we used a chi-square ($X^2$) test for equal proportions. The proportion of skip preferring cells was determined as those with an average net firing rate greater than 0 and was compared against a null hypothesis of 50% of cell in the respective mPFC subregion. Note that because accept vs skip is a binary choice the proportion of accept preferring cells is the direct inverse with identical statistical results.

Correlation between average firing rates and offer delay (*Figure 6*) and linger time (*Figure 7*) were performed after first averaging firing rate PETHs for each rat independently. For each rat, PETHs were generated for each behavioral bin yielding up to 48 firing rate profiles (8 rats x 6 bins). For comparing offer delay firing in the WZ behavior was grouped into evenly spaced bins according to the offer presented in the OZ. For comparisons to linger time we used bins composed of percent of the total linger time. In each session the 1st and 99th percentiles of linger time were taken as the session minimum and maximum. For each visit to the RZ, the total linger time was computed as the relative percentage between this minimum and maximum. Using this method, included linger times between 0 and 30% accounted for 87% of all RZ visits. From these PETH profiles average firing rates were computed in four time windows for each analysis. For the WZ, time bins were 0–1 s before WZ entry (Pre WZ), 0–3 s after WZ entry (Early WZ), 0–3 s before WZ exit (Late WZ), and 0–1 s after WZ exit (Post WZ). For the RZ, time bins were the 3 bins prior to RZ entry (Pre RZ), the first 10 bins of the RZ (Early RZ), the second 10 bins of the RZ (Late RZ), and the 3 bins after RZ exit (Post RZ).

All correlations between sets of values used Pearson correlations and significance assessed at $\alpha=0.05$. Unless otherwise indicated, significance was assessed using two-sided tests and an $\alpha=0.05$.

## Acknowledgements

We thank K Seeland, C Boldt, A Sheehan, and C Bogner for technical assistance as well as members of the Redish lab for useful discussion. We also thank S Heilbronner for discussion and guidance regarding prefrontal anatomy, and the G Buszáki lab for guidance on silicon probe recordings. This work was supported by NIH grants R01-MH112688 and T32-MH115886.

## Additional information

### Funding

| Funder | Grant reference number | Author |
|---|---|---|
| National Institute of Mental Health | T32 MH115886 | Geoffrey W Diehl |
| National Institute of Mental Health | R01 MH112688 | A David Redish |

The funders had no role in study design, data collection and interpretation, or the decision to submit the work for publication.

### Author contributions

Geoffrey W Diehl, Conceptualization, Data curation, Formal analysis, Validation, Investigation, Visualization, Methodology, Writing – original draft, Writing – review and editing, Data collection; A David Redish, Conceptualization, Supervision, Funding acquisition, Investigation, Writing – original draft, Project administration, Writing – review and editing

### Author ORCIDs

A David Redish http://orcid.org/0000-0003-3644-9072

### Ethics

This study was performed in strict accordance with the recommendations in the Guide for the Care and Use of Laboratory Animals of the National Institutes of Health. All animal handling and experimental protocols were approved by the Institutional Animal Care and Use Committee (IACUC) at the University of Minnesota (protocol #1910-37469A).

### Decision letter and Author response

Decision letter https://doi.org/10.7554/eLife.82833.sa1
Author response https://doi.org/10.7554/eLife.82833.sa2

## Additional files

### Supplementary files
• MDAR checklist

### Data availability

Data and supporting code has been uploaded to OSF. 2022 RedishLab: medial prefrontal subregions recorded from rats on restaurant row. https://osf.io/s5xqm/.

The following dataset was generated:

| Author(s) | Year | Dataset title | Dataset URL | Database and Identifier |
|-----------|------|---------------|-------------|------------------------|
| Diehl GW, Redish AD | 2022 | RedishLab: medial prefrontal subregions recorded from rats on restaurant row | https://osf.io/s5xqm/ | Open Science Framework, s5xqm |

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
