## [Editor Report]

In this study, Diehl and Redish present a novel and fundamental account of functional variability in the rodent medial prefrontal cortex, in which the dorsal regions encode decision-related variables and the ventral regions encode variables linked to motivation. The study's experimental design is excellent, the analyses are appropriate, and the conclusions are based on compelling evidence. The suggestion of functional subdivisions in the prelimbic area is particularly provocative, and this conclusion, along with the data supporting it, will be of broad interest to those who study the anatomy and function of the rodent medial prefrontal cortex.

---

## [Decision Letter]

**Decision letter after peer review:**

Thank you for submitting your article "Differential processing of decision information in subregions of rodent medial prefrontal cortex" for consideration by *eLife*. Your article has been reviewed by 3 peer reviewers, and the evaluation has been overseen by a Reviewing Editor and Kate Wassum as the Senior Editor. The following individual involved in the review of your submission has agreed to reveal their identity: James Hyman (Reviewer #1).

Essential revisions:

In your revised submission, please address the following points, which are considered essential revisions. Full comments from each reviewer are appended to this email for your reference and include further discussion of these points, as well as additional suggestions for improving the manuscript. Please provide a response to all points along with your resubmission.

1. Reviewers felt that the framing of the study and results should be tempered somewhat. For instance, the hypotheses in Figure 8 are more of a schematic illustration showing the extremes of a continuum of potential differentiation within mPFC. While these limit cases can serve the purpose of defining a space of possibilities, it should be clearer that these are oversimplified for illustration and, based on extant data, the actual results are much more likely (and in fact do) to fall in the middle of this continuum. For instance, the results appear consistent with a gradient of coding task phase versus session time, so that terms like "fundamental dichotomy" seem too strong. Importantly, tempering such claims would not detract from the novelty and importance of the results, and would simply be a more accurate representation of the findings.

2. There were questions about how the unequal sampling of ACC compared to other regions impacted the results. Reviewers suggest an additional analysis to show that the findings are robust, for instance randomly sampling a similar number of units from the other areas to determine how sensitive the results are to sample size, or taking a dimensionality reduction approach to finding the number of dimensions needed to explain significant variance in each population.

3. In this task, there are correlations between choices and motor behavior. Given that correlates of movement are found in ACC, it should be shown that the choice signals reported here are not simply motor correlates (e.g., orienting movements). One suggestion is to assess whether these signals are observed for movements outside of the task, or only really reflect a task-related decision.

4. In several cases, the authors correlate neural firing rates are correlated against behavioral variables, but the authors show that some of these variables are correlated with each other, which might result in spurious correlations between some variables and neural activity. Reviewers requested an analysis that parses these correlations, for instance, a regression-based approach that takes into account the correlations between variables.

5. Reviewers requested clarification of each of the following:

– The shuffling procedure for transfer entropy. Please clarify how the timebins were repeatedly shuffled to produce a single value of TE.

– For figure 4, are the responses time-warped so they are all the same duration?

– For figure 2D, please clarify the interpretation, specifically what the changing values here mean.

– Please clarify how the data in figure 3B is related to that in figure 3C. It seems that they should correspond, but it is not clear how.

– In figure 6D, dPL is colored yellow for re-evaluation, but please clarify what the other colors in the other regions mean.

6. Results should be discussed in relation to Sul JH et al., Neuron, 2010.

7. It would be interesting to explore how the results of transfer entropy and other analyses change over the course of the session since the authors conclude that more ventral areas are more affected by motivational factors that are likely to change throughout the session.

*Reviewer #1 (Recommendations for the authors):*

Our main concern is with the limited number of ACC cells in comparison to the other areas. The authors need to take multiple steps to address this potentially fatal flaw. First, they need to more forcefully point out that these ACC findings are entirely consistent with a plethora of previous work. This should be pointed out each time they present a new conclusion about ACC (even within the Results). Next, the authors should take steps to level the playing field for between-area comparisons by randomly sampling equivalent numbers of cells from each area and then looking at the results. Such bootstrapping should be repeated and then the results should be reported either as an additional control or as a replacement for the current results. Lastly, I think the authors need to hedge every conclusion they make about ACC with the possibility that with such a small sample, there could be sampling bias.

We're curious how transfer entropy changes and other analyses change over the course of the session. The authors conclude that more ventral areas are more affected by motivational factors, but motivation is not consistent throughout the session. Does this affect transfer entropy? Are values consistent from the beginning to the end of the session? What about for functional correlates? Shouldn't changing motivation affect population dynamics in these ventral areas?

How do the authors account for the behavioral differences that are present between Earn vs. Quit decisions? In one case the animal is walking away and the other the animal consumes the reward, both of which could greatly affect spiking.

*Reviewer #2 (Recommendations for the authors):*

– Appreciate the display of histology for all the probe tracks in Supplementary Figure 2, which supports the exclusion of 2 of the recordings.

– "Notably, the proportion of cells with a significant relation to behavior was not equivalent across all behavioral variables, but instead were biased towards those variables that intuitively would seem more critical for succeeding in the task" – more evidence is needed to support this statement. What are the variables needed for success, and how would one rank them, and are there statistically significant differences?

– Figure 2D – not sure how to interpret this – "Proportion of mPFC cell population that was significantly correlated to both of a pair of behavioral variables", I would have expected the identity diagonal (session time and sessions time, task phase, and task phase, etc.) to equal to one? What do the changing values here mean?

– Figure 3C, I don't understand how this plot is constructed. One can see at DV = 4.2, the matrix has many red pixels and therefore the value in Figure 3C is correspondingly high. However, at around DV = 2.5, most of the pixels in the identity diagonal and even adjacent areas are blue. How does this DV yield the highest norm? TE of all in Figure 3C?

– Notwithstanding the question regarding the quantification, the result that one could subdivide the mPFC based on the functional measure is quite interesting.

– The results show a gradient of coding in terms of frontal cortical regions and task phase vs session time. The language used in the text, e.g., "implies a fundamental dichotomy within the mPFC", seems too strong.

– Relatedly, the closest relevant prior study to the current experiment is probably Sul JH et al., Neuron, 2010, which is omitted from the reference list. Sul and colleagues compared the neural correlates in various frontal cortical regions during a two-armed bandit task. Their findings of ACC involving the manifestation of the animal's choice, whereas OFC and mPFC convey signals related to the animal's past over multiple trials (over a long timescale, as implied by the analyses in this study relating to session time) are definitely related to the current study. The authors should discuss the similarities and differences between the studies.

– Figure 6B – all of the regions have residual signals related to the WZ choice after the WZ exit. In my opinion, the results indicate a gradient of functions, rather than what the title suggests which is more definitive "Dorsal PL firing encodes re-evaluate quit decisions", when in fact these signals are also present in other frontal cortical regions.

– The "hypotheses" listed in Figure 8A-C are simplistic. Taken literally, a fully distributed system where "everything is everywhere" and equally distributed where anatomy provides no information just does not occur anywhere in the brain. The other situation of subregions where functions are restricted to one part of mPFC is also improbable given the recurrent and long-range connections across regions and the fact that lesions have not provided definitive effects on behavioral function. Overall I do not think fitting the results into this specific framework is helpful for interpreting the results.

– Figure 6D – for this cartoon, for dPL is colored yellow for re-evaluation, then what do the other colors used for the other regions mean? What do these colors signify? If you truly believe in subregions, shouldn't the other areas be white or blank?

*Reviewer #3 (Recommendations for the authors):*

I have suggestions that I will outline below that I hope will strengthen the manuscript, including the weaknesses I've described.

1. The authors report that ACC strongly encodes the upcoming decision of accepting or skipping the offer. However, motor variables are also correlated with the decision, as the decision to accept the offer and enter the wait zone requires different orienting movements than skipping and continuing along the track. Previous recordings from more posterior regions of the rat cingulate cortex have found encoding of contralateral vs. ipsilateral orienting movements (e.g., Erlich et al., 2011, Neuron). Can the authors rule out the possibility that encoding of the upcoming choice in ACC does not reflect different orienting movements of the animal (e.g., ipsi vs. contra to the recording site)?

2. In several cases, the authors correlate neural firing rates against behavioral variables. The authors show that some of these variables are correlated with each other, which might result in spurious correlations between some variables and neural activity. The authors should consider regressing the different variables against neural activity in a single regression model and examining the number of significant coefficients.

3. For figure 4, are the responses time-warped so they are all the same duration?

4. Also for figure 4, the fact that the words "Accept/Skip" are by the color bar that reflects the firing rates is confusing – it makes it look as though the color of the heatmats relates to the animal's choice, as opposed to neural activity. Perhaps put accept/skip-preferring? Or remove?

5. Figure 4C: I didn't find this depiction to be particularly helpful or easy to read. Perhaps displaying the data as bar plots would make it easier to compare the magnitude of responses both within an area and between areas.

6. For figure 6, the interpretation and title of that figure are that dPL encodes re-evaluation, but the encoding is also present in vPL and IL (at least in terms of the correlation).

7. In the Methods description of transfer entropy, I was confused about the shuffling procedure. It sounds like they shifted the spike times of cell X by one time bin, but then they said they did that 30 times. But for a given vector X and Y, there should be a single value of TE, no? Do they mean that they shifted the spike times of cell X by some random time bin, x 30? Or by 1-31 time bins? Please clarify the shuffling procedure.

---

## [Author Response]

Essential revisions:In your revised submission, please address the following points, which are considered essential revisions. Full comments from each reviewer are appended to this email for your reference and include further discussion of these points, as well as additional suggestions for improving the manuscript. Please provide a response to all points along with your resubmission.

We thank the reviewer and editors for their useful input on our manuscript. We have taken the below steps to address their concerns and improve the manuscript. All changes to the manuscript are identified with “track changes”. We have also added 10 additional supplemental figures.

1. Reviewers felt that the framing of the study and results should be tempered somewhat. For instance, the hypotheses in Figure 8 are more of a schematic illustration showing the extremes of a continuum of potential differentiation within mPFC. While these limit cases can serve the purpose of defining a space of possibilities, it should be clearer that these are oversimplified for illustration and, based on extant data, the actual results are much more likely (and in fact do) to fall in the middle of this continuum. For instance, the results appear consistent with a gradient of coding task phase versus session time, so that terms like "fundamental dichotomy" seem too strong. Importantly, tempering such claims would not detract from the novelty and importance of the results, and would simply be a more accurate representation of the findings.

We have now edited text throughout the manuscript to temper any discussion of “fundamental dichotomies” and directly associating any single prefrontal subregion with any particular functional characteristic. We now discuss differences between subregions as functional biases and the presence of a gradient along the dorso-ventral axis. We have also modified our discussion of the hypotheses originally from Figure 8, both in the discussion text and by altering the content of the figure to more appropriately match our results (other aspects of this figure were mentioned elsewhere by reviewers).

2. There were questions about how the unequal sampling of ACC compared to other regions impacted the results. Reviewers suggest an additional analysis to show that the findings are robust, for instance randomly sampling a similar number of units from the other areas to determine how sensitive the results are to sample size, or taking a dimensionality reduction approach to finding the number of dimensions needed to explain significant variance in each population.

We have now taken additional steps to address our limited sampling in ACC. First, we have repeated our TE analysis after bootstrapping the samples from the other three subregions to yield comparable numbers of cells. We have also replicated our OZ, WZ, and RZ figures (Figure 5-7) in supplements using subsampled data from the dPL, vPL, and IL subregions. Across all cases our results were robust to down-sampling. We also discuss at each stage of the Results section that our sampling of ACC may be susceptible to biases.

3. In this task, there are correlations between choices and motor behavior. Given that correlates of movement are found in ACC, it should be shown that the choice signals reported here are not simply motor correlates (e.g., orienting movements). One suggestion is to assess whether these signals are observed for movements outside of the task, or only really reflect a task-related decision.

First, we now include a supplemental figure where we examine OZ activity split by hemisphere location of each recorded cell to leverage ipsilateral vs contralateral direction of motion, as suggested by Reviewer 3. This found largely balanced preference for ipsi and contralateral movement which would be more in line with a role in decision making as opposed to motor action based on the work by Erlich et al. Second, we include a supplemental figure examining net firing rate differences in the WZ between quit decisions (movement) and earn decisions (no movement). Again, firing rate biases were balanced between the options indicative of choice coding. Finally, we now include a paragraph in our discussion highlighting the intrinsic issues associated with parsing between choice and motor behavior in free choice tasks and the steps that we have taken here to address this confound.

4. In several cases, the authors correlate neural firing rates are correlated against behavioral variables, but the authors show that some of these variables are correlated with each other, which might result in spurious correlations between some variables and neural activity. Reviewers requested an analysis that parses these correlations, for instance, a regression-based approach that takes into account the correlations between variables.

We have now included two additional analyses in which we perform a stepwise regression to progressively remove explanatory factors when comparing to neural firing. In the first case, we performed the stepwise regression where we input variables into the regression in an experimenter assigned ordering to maintain the same sequence across all cells. Order was taken from the overall prevalence in single variable correlation from the full data set; Figure 2C. In the second case, we used the builtin matlab function “stepwisefit” which allowed variable input ordering to be determined for each cell independently according to explanatory power.

5. Reviewers requested clarification of each of the following:– The shuffling procedure for transfer entropy. Please clarify how the timebins were repeatedly shuffled to produce a single value of TE.

We have updated the text in the methods section to clarify this method.

– For figure 4, are the responses time-warped so they are all the same duration?

We believe that the reviewers are referring here to figure 5, discussion of the offer zone activity. They are correct that responses are time-warped so that they are all the same duration. We have added a comment in the methods section (Normalization of Offer and Reward Zone time) that identifies that this method applies to figures 5 and 7D. We have also added comments to this respect in the legends of the corresponding figures.

– For figure 2D, please clarify the interpretation, specifically what the changing values here mean.

The diagonal in figure 2D represented the proportion of cells that were significantly related to the single behavioral variable (i.e. proportion of cells that are correlated to Session Time). This is a different interpretation than the off-diagonal bins in this plot (Behavior 1 *and* Behavior 2) and we see how this caused confusion. We have now changed the display of this data such that the separate information that used to be on the diagonal is presented alone as a separate plot.

– Please clarify how the data in figure 3B is related to that in figure 3C. It seems that they should correspond, but it is not clear how.

The reviewers are correct that the data in 3B and 3C correspond. First, the quantification in 3C includes bins that are anatomically within 200um of corresponding depth, meaning that bins slightly off-diagonal will also contribute. We have sought to clarify this in the display of 3B and the associated supplementary figure by expanding the illustration of the diagonal to match the quantification. Second, the lack of obvious clarity in the correspondence likely relates to our lower sampling in the ACC subregion. As noted elsewhere, while this limited sampling of ACC is unfortunate, we believe that our data are still able to provide an accurate picture of ACC.

– In figure 6D, dPL is colored yellow for re-evaluation, but please clarify what the other colors in the other regions mean.

We believe that this is referring to Figure 8D. The coloring of dPL matches the color scheme we adopted throughout the manuscript as do the labeling of the other mPFC subregions. We have now reworked this figure based on other reviewer comments, included additional text in the figure and its legend, and believe that the presentation is now more clear.

6. Results should be discussed in relation to Sul JH et al., Neuron, 2010.

We thank the reviewers for alerting us to this oversight. We now cite this paper where appropriate throughout the manuscript and specifically include a paragraph on this work in the discussion.

7. It would be interesting to explore how the results of transfer entropy and other analyses change over the course of the session since the authors conclude that more ventral areas are more affected by motivational factors that are likely to change throughout the session.

We now include a supplementary figure that examines TE and the main functional effects in the OZ, WZ, and RZ over the course of the session.

Reviewer #1 (Recommendations for the authors):Our main concern is with the limited number of ACC cells in comparison to the other areas. The authors need to take multiple steps to address this potentially fatal flaw. First, they need to more forcefully point out that these ACC findings are entirely consistent with a plethora of previous work. This should be pointed out each time they present a new conclusion about ACC (even within the Results). Next, the authors should take steps to level the playing field for between-area comparisons by randomly sampling equivalent numbers of cells from each area and then looking at the results. Such bootstrapping should be repeated and then the results should be reported either as an additional control or as a replacement for the current results. Lastly, I think the authors need to hedge every conclusion they make about ACC with the possibility that with such a small sample, there could be sampling bias.

We have now performed additional analyses in which we subsample the data from the other mPFC subregions to match the sampling that we have from ACC. These are included as new supplementary figures. Importantly, the qualitative results from all analyses hold when subsampling the other subregions. This suggests that while some of our ACC results may be underpowered, they likely reflect an accurate picture of the functional properties of this prefrontal subregion. As suggested, we also discuss our results for ACC as consistent with the broader literature and note the potential sampling biases that may exist within our ACC data set.

We're curious how transfer entropy changes and other analyses change over the course of the session. The authors conclude that more ventral areas are more affected by motivational factors, but motivation is not consistent throughout the session. Does this affect transfer entropy? Are values consistent from the beginning to the end of the session? What about for functional correlates? Shouldn't changing motivation affect population dynamics in these ventral areas?

We now include an additional supplementary figure that examines TE analysis and the primary results in the OZ, WZ, and RZ over the course of the session.

How do the authors account for the behavioral differences that are present between Earn vs. Quit decisions? In one case the animal is walking away and the other the animal consumes the reward, both of which could greatly affect spiking.

This is a good point. We now include a supplementary figure in which we examine the net firing rate changes between Quit and Earn decisions. If motor actions were driving neural activity we would expect increases in firing rate only for Quit decisions when rats engage a motor action. Instead we see a balance across the population of cells that prefer quits and cells that prefer earns, suggesting a decision related drive. We also now include a comparison of the neural activity during quit decisions and the neural activity when rats leave the RZ after consuming a reward as these will be the closest motor comparison. This suggests that while there may be some aspect of motor action driving mPFC activity, especially in the dorsal areas, it is likely a more minor factor. We now also include a paragraph in the discussion on the confound of decision and motor action.

Reviewer #2 (Recommendations for the authors):– Appreciate the display of histology for all the probe tracks in Supplementary Figure 2, which supports the exclusion of 2 of the recordings.

Thank you.

– "Notably, the proportion of cells with a significant relation to behavior was not equivalent across all behavioral variables, but instead were biased towards those variables that intuitively would seem more critical for succeeding in the task" – more evidence is needed to support this statement. What are the variables needed for success, and how would one rank them, and are there statistically significant differences?

We no longer discuss behavioral variables as being critical for succeeding in the task and only identify that variables were not all represented equally. By the very nature of the RRow task, rats are free to utilize whatever strategies they choose to solve the task. They are simply limited to 1 hr of time on the track. As such, we have removed this comment and thank the reviewer for raising this important point.

– Figure 2D – not sure how to interpret this – "Proportion of mPFC cell population that was significantly correlated to both of a pair of behavioral variables", I would have expected the identity diagonal (session time and sessions time, task phase, and task phase, etc.) to equal to one? What do the changing values here mean?

The diagonal in figure 2D represents the proportion of cells that were significantly related to the single behavioral variable (i.e. proportion of cells that are correlated to Session Time). This is a different interpretation than the off-diagonal bins in this plot (Behavior 1 and Behavior 2) and we see how this caused confusion. We have now changed the display of this data such that the information on the diagonal is presented in its own figure panel.

– Figure 3C, I don't understand how this plot is constructed. One can see at DV = 4.2, the matrix has many red pixels and therefore the value in Figure 3C is correspondingly high. However, at around DV = 2.5, most of the pixels in the identity diagonal and even adjacent areas are blue. How does this DV yield the highest norm? TE of all in Figure 3C?

First, the quantification in 3C includes bins that are anatomically within 200um of corresponding depth, meaning that bins slightly off-diagonal will also contribute. We have sought to clarify this in the display of 3B by expanding the illustration of the diagonal to match the quantification. Second, the lack of obvious clarity in the correspondence likely relates to our lower sampling in the ACC subregion. As noted elsewhere, while this limited sampling of ACC is unfortunate, we believe that our data are still able to provide an accurate picture of ACC.

– Notwithstanding the question regarding the quantification, the result that one could subdivide the mPFC based on the functional measure is quite interesting.

Thank you, we agree.

– The results show a gradient of coding in terms of frontal cortical regions and task phase vs session time. The language used in the text, e.g., "implies a fundamental dichotomy within the mPFC", seems too strong.

We have removed any discussion of prefrontal as a “fundamental dichotomy” between dorsal and ventral and now discuss our findings with regards to functional biases and gradients along the medial wall.

– Relatedly, the closest relevant prior study to the current experiment is probably Sul JH et al., Neuron, 2010, which is omitted from the reference list. Sul and colleagues compared the neural correlates in various frontal cortical regions during a two-armed bandit task. Their findings of ACC involving the manifestation of the animal's choice, whereas OFC and mPFC convey signals related to the animal's past over multiple trials (over a long timescale, as implied by the analyses in this study relating to session time) are definitely related to the current study. The authors should discuss the similarities and differences between the studies.

We thank the reviewer for alerting us to this oversight. We now cite this paper where appropriate throughout the manuscript and specifically include a paragraph on this work in the discussion.

– Figure 6B – all of the regions have residual signals related to the WZ choice after the WZ exit. In my opinion, the results indicate a gradient of functions, rather than what the title suggests which is more definitive "Dorsal PL firing encodes re-evaluate quit decisions", when in fact these signals are also present in other frontal cortical regions.

We agree. We have adapted the title of this figure and relevant heading in the Results section to not indicate that dPL is the only subregion coding quit decisions. As noted elsewhere we now discuss our results as indicative of a functional gradient.

– The "hypotheses" listed in Figure 8A-C are simplistic. Taken literally, a fully distributed system where "everything is everywhere" and equally distributed where anatomy provides no information just does not occur anywhere in the brain. The other situation of subregions where functions are restricted to one part of mPFC is also improbable given the recurrent and long-range connections across regions and the fact that lesions have not provided definitive effects on behavioral function. Overall I do not think fitting the results into this specific framework is helpful for interpreting the results.

We have now adapted the relevant portions of the Discussion section and made changes to figure 8 to address this concern.

– Figure 6D – for this cartoon, for dPL is colored yellow for re-evaluation, then what do the other colors used for the other regions mean? What do these colors signify? If you truly believe in subregions, shouldn't the other areas be white or blank?

We believe that this is referring to Figure 8D. The coloring of dPL matches the color scheme we adopted throughout the manuscript as does the labeling of the other mPFC subregions. That said, we have now reworked this figure and hope that the new iteration provides a clearer illustration, and addresses some of the concerns raised in the above point.

Reviewer #3 (Recommendations for the authors):I have suggestions that I will outline below that I hope will strengthen the manuscript, including the weaknesses I've described.1. The authors report that ACC strongly encodes the upcoming decision of accepting or skipping the offer. However, motor variables are also correlated with the decision, as the decision to accept the offer and enter the wait zone requires different orienting movements than skipping and continuing along the track. Previous recordings from more posterior regions of the rat cingulate cortex have found encoding of contralateral vs. ipsilateral orienting movements (e.g., Erlich et al., 2011, Neuron). Can the authors rule out the possibility that encoding of the upcoming choice in ACC does not reflect different orienting movements of the animal (e.g., ipsi vs. contra to the recording site)?

We now include a supplemental figure on OZ activity to try and address this concern. Briefly, we employ the analysis suggested here (and thank the reviewer for the suggestion) and examine firing split by hemisphere to leverage ipsilateral vs contralateral direction of motion. We also attempt to address a related concern of decision vs motion in the WZ regarding quit/earn decisions in two additional supplemental figures. Finally, we now include an additional paragraph in the discussion touching on the confound of decision vs motor in free moving tasks and highlight the steps that we have taken here.

2. In several cases, the authors correlate neural firing rates against behavioral variables. The authors show that some of these variables are correlated with each other, which might result in spurious correlations between some variables and neural activity. The authors should consider regressing the different variables against neural activity in a single regression model and examining the number of significant coefficients.

We now include an additional supplemental figure in which we perform stepwise regression analyses.

3. For figure 4, are the responses time-warped so they are all the same duration?

We believe that the reviewer is referring here to figure 5, discussion of the offer zone activity. They are correct that responses are time-warped so that they are all the same duration. We have added a comment in the methods section (Normalization of Offer and Reward Zone time) that identifies that this method applies to figures 5 and 7D. We have also added comments to this respect in the legends of the corresponding figures.

4. Also for figure 4, the fact that the words "Accept/Skip" are by the color bar that reflects the firing rates is confusing – it makes it look as though the color of the heatmats relates to the animal's choice, as opposed to neural activity. Perhaps put accept/skip-preferring? Or remove?

We have adjusted the display of this figure (5B) accordingly.

5. Figure 4C: I didn't find this depiction to be particularly helpful or easy to read. Perhaps displaying the data as bar plots would make it easier to compare the magnitude of responses both within an area and between areas.

We now show these data as bar plots.

6. For figure 6, the interpretation and title of that figure are that dPL encodes re-evaluation, but the encoding is also present in vPL and IL (at least in terms of the correlation).

We have modified the title of this figure and the corresponding results subheading accordingly.

7. In the Methods description of transfer entropy, I was confused about the shuffling procedure. It sounds like they shifted the spike times of cell X by one time bin, but then they said they did that 30 times. But for a given vector X and Y, there should be a single value of TE, no? Do they mean that they shifted the spike times of cell X by some random time bin, x 30? Or by 1-31 time bins? Please clarify the shuffling procedure.

We have added additional text in the methods section outlining the shuffling procedure for TE calculations.